# A Triple Gene-Deleted Pseudorabies Virus-Vectored Subunit PCV2b and CSFV Vaccine Protects Pigs against PCV2b Challenge and Induces Serum Neutralizing Antibody Response against CSFV

**DOI:** 10.3390/vaccines10020305

**Published:** 2022-02-16

**Authors:** Selvaraj Pavulraj, Katrin Pannhorst, Rhett W. Stout, Daniel B. Paulsen, Mariano Carossino, Denise Meyer, Paul Becher, Shafiqul I. Chowdhury

**Affiliations:** 1Department of Pathobiological Sciences and Louisiana Animal Disease Diagnostic Laboratory, School of Veterinary Medicine, Louisiana State University, Baton Rouge, LA 70803, USA; pselvaraj1@lsu.edu (S.P.); katrin.pannhorst@fli.de (K.P.); rstout1@lsu.edu (R.W.S.); dpauls1@lsu.edu (D.B.P.); mcarossino1@lsu.edu (M.C.); 2Institute of Virology, University of Veterinary Medicine Hannover, 30559 Hannover, Germany; denise.meyer@tiho-hannover.de (D.M.); paul.becher@tiho-hannover.de (P.B.)

**Keywords:** pseudorabies virus, triple mutant, vectored vaccine, PCV2 capsid, CSFV, glycoproteins E2 and Erns, PRV trivalent vaccine, vaccine efficacy, granulocytic monocyte-colony stimulating factor (GM-CSF), pig

## Abstract

Porcine circovirus type 2 (PCV2) is endemic worldwide. PCV2 causes immunosuppressive infection. Co-infection of pigs with other swine viruses, such as pseudorabies virus (PRV) and classical swine fever virus (CSFV), have fatal outcomes, causing the swine industry significant economic losses in many if not all pig-producing countries. Currently available inactivated/modified-live/vectored vaccines against PCV2/CSFV/PRV have safety and efficacy limitations. To address these shortcomings, we have constructed a triple gene (thymidine kinase, glycoprotein E [gE], and gG)-deleted (PRVtmv) vaccine vector expressing chimeric PCV2b-capsid, CSFV-E2, and chimeric E^rns^-fused with bovine granulocytic monocyte-colony stimulating factor (E^rns^-GM-CSF), designated as PRVtmv+, a trivalent vaccine. Here we compared this vaccine’s immunogenicity and protective efficacy in pigs against wild-type PCV2b challenge with that of the inactivated Zoetis Fostera Gold PCV commercial vaccine. The live PRVtmv+ prototype trivalent subunit vaccine is safe and highly attenuated in pigs. Based on PCV2b-specific neutralizing antibody titers, viremia, viral load in lymphoid tissues, fecal-virus shedding, and leukocyte/lymphocyte count, the PRVtmv+ yielded better protection for vaccinated pigs than the commercial vaccine after the PCV2b challenge. Additionally, the PRVtmv+ vaccinated pigs generated low to moderate levels of CSFV-specific neutralizing antibodies.

## 1. Introduction

Porcine circovirus type 2 (PCV2) is endemic in most, if not all, intensive pig farms worldwide, including the United States, as evidenced by a remarkably high seroprevalence [1]. The virus was first isolated from pigs with the post-weaning multisystemic wasting syndrome (PMWS) in the 1990s [2]. PCV2 infection and replication in the lymphoid tissues in the host destroys the architecture of lymphoid follicles, leading to lymphoid depletion and histiocytic replacement of lymphoid tissues. PCV2 also interacts with host cellular factors and modulates the host immune response. Consequently, PCV2 infection can lead to immunosuppression and cytokine imbalance [3]. Although PCV2 alone causes only sub-clinical disease, because of its immunosuppressive nature, PCV2 co-infection with *Mycoplasma* spp., parvovirus, or porcine reproductive and respiratory syndrome virus (PRRSV) may cause the clinical disease designated as PMWS [4]. Additionally, the preexisting PCV2 infection decreases the efficacy of vaccines. Additionally, coinfection of PCV2 with swine influenza virus (SIV), classical swine fever virus (CSFV), and pseudorabies virus (PRV) also increases the pathogenicity of these infections [5]. Consequently, PCV2-associated diseases pose a significant problem for the pig industry worldwide [6] and cause substantial economic losses in many pig-producing countries, including the USA.

Classical swine fever (CSF) is a highly contagious disease of swine and is still endemic in many countries, including China, where half of the world’s pig population is located. CSF is regarded as one of the significant problems in pig-producing countries, i.e., China and other Southeastern Asian, East European, and some European Union (EU) countries [7]. Routine prophylactic vaccination is practiced in many affected regions (e.g., China, Russia, and Southeast Asia) to prevent CSF losses. Several countries in Europe, the US, Canada, New Zealand, and Australia are currently declared CSF-free [8,9]. However, there is always a risk of reintroducing CSF with disastrous economic losses. Several EU countries experienced a series of CSFV outbreaks in the 1980s and 1990s. The costs of the epizootic in the Netherlands alone exceeded $2 billion [10]. In China, the live attenuated CSFV vaccine is suspected to help cause and maintain the CSFV endemic status [11].

Pseudorabies, or Aujeszky’s disease (AD), caused by PRV, is another economically significant viral disease of pigs in many regions, especially in China and many developing countries [12]. In domestic pigs, strict control measures and the use of gene-deleted marker vaccines resulted in the elimination of PRV infections in many parts of Europe and North America. However, PRV remains more widely distributed in free-roaming feral pigs and wild boar populations both in Europe and North America. Thus, there is a constant risk of spillover of PRV infection from wild pig populations to domestic animals [13]. PRV modified live vaccines (MLVs), such as the Bartha-K61 strain, have successfully controlled the disease in many countries, including China [14]. However, since late 2011, a highly pathogenic, economically significant PRV strain has emerged in many Bartha-K61-vaccinated swine herds in many regions of China [15,16].

Since PCV2, CSFV, and PRV are endemic, and co-infections are common in intensive pig farms in China [17], the severity and fatality of the disease and the emergence of a highly pathogenic PRV strain could have been the result of complex epidemiological interactions between the PRV and co-infecting virus(es) [18]. Both CSFV and PRV have been eradicated in the US and many European countries. However, both the viruses are maintained in the wild pigs in some, if not all, European countries and PRV in feral swine in the US [13]. Additionally, CSFV is endemic in some South and Central American countries [19]. CSFV reemerged in Cuba in 1993 and has since spread to Haiti (1996) and the Dominican Republic (1997) [20].

The PCV2 has a single-stranded DNA genome that comprises 1767 or 1768 nucleotides (nt) and is predicted to have 11 potential open reading frames (ORF1–11) [21]. Despite being the smallest genome among other DNA viruses, the virus has the highest evolution rate [22]. Therefore, PCV2 continues to evolve, and currently there are five different genotypes, PCV2a to PCV2e. During the mid-2000s, pig populations were mainly infected with the PCV2b genotype. Since 2012, however, the dominant PCV2b genotype was replaced by PCV2d genotype in most pig production countries, including North America and China. Nevertheless, both genotypes are antigenically related and produce cross-neutralizing antibodies [23]. Previous studies have demonstrated that ORF2 encodes a 27.8 kDa capsid (Cap) protein. PCV2 Cap is highly immunogenic and produces neutralizing antibodies in pigs [24]. Additionally, neutralizing pig sera recognized the PCV2 Cap [25]. Because PCV2 has the highest mutation rates among the DNA viruses, only subunit protein or inactivated vaccines are used and are not adequately protective [22].

CSFV is enveloped and has a positive-sense, single-stranded, 12.5 kb RNA genome [26]. The genome contains a single ORF that encodes a polyprotein composed of 3898 amino acids (aa) that is cleaved proteolytically by viral and cellular proteases to yield up to 12 products (NH2-NproC-E^rns^-E1-E2-p7-NS2-NS3-NS4A-NS4B-NS5A-NS5B-COOH) [27]. There are four virion-associated structural proteins, C (capsid) and E^rns^, E1, and E2 (envelope proteins). Although E^rns^ is associated loosely with the envelope, the protein is also secreted [28]. E2 has been implicated, along with E^rns^ and E1, in viral adsorption to host cells [29]. E2 is essential for CSFV replication, as virus mutants containing partial or complete deletions of the E2 gene are nonviable [30]. Both E2 and E^rns^ elicit neutralizing antibodies and induce protective immunity independently [31]. Countries free of CSFV, including the EU countries, do not apply vaccination to their national herds despite currently available CSFV- MLV, which confers adequate, rapid, and solid immune protection and limits the severity of the diseases in vaccinated animals [32]. Sporadic outbreaks and endemic forms of CSF in China and Korea are believed to have occurred due to the extensive use of MLV [11,33].

The PRV genome is approximately 145 kb and is composed of numerous essential and non-essential genes for replication in vitro in epithelial cells. Among the non-essential genes, thymidine kinase (TK), envelope glycoprotein E (gE), and gG deletions individually or collectively do not affect the in vitro replication efficiency of the virus in cell culture. However, the TK-, gE-, or gG-deletion individually or simultaneously reduces the virulence of the virus in pigs [12,34,35]. Although TK is non-essential for viral replication in vitro in epithelial cells, the TK-deleted virus has defective replication in neurons, which are post-mitotic and lack TK [36].

PRV gE-deleted virus replicates efficiently both in cell culture in vitro and in nasal epithelium of pigs in vivo; however, the gE-deleted virus is highly attenuated in pigs and does not produce clinical signs following intranasal (IN) infection [34,35]. PRV gE-deleted-vaccinated pigs can be differentiated from the infected pigs (DIVA). Therefore, PRV gE-deleted vaccine is used for eradication efforts in many countries [12].

PRV envelope glycoprotein gG is a viral chemokine binding protein [37]. The binding of PRV-gG to chemokines results in interference with chemokine-mediated lymphocyte, neutrophil, and monocyte migration to the site of virus infection, thereby PRV gG has a role in evading the host’s immune response [37]. In a field trial, a gE and gG deleted virus was more immunogenic when compared with only a gE-deleted virus [38].

We have constructed a triple gene-deleted PRV vector (PRVtmv), in which we have deleted the TK, and envelope glycoproteins gG and gE genes. In addition, the chimeric PCV2b capsid (Cap), CSFV E2, and a chimeric CSFV E^rns^-GM-CSF (E^rns^ fused with granulocyte-macrophage colony-stimulating factor; GM-CSF) genes were inserted in the TK-, gE-, and gG-deletion loci, respectively, resulting in a PRVtmv vectored PCV2b Cap, CSFV E2, and E^rns^-GM-CSF expressing virus (referred to hereafter as PRVtmv+). In this study, the PRVtmv+ in vitro characterization, pathogenicity in pigs, and protective serum neutralizing antibody titers against PCV2b before and after the PCV2b challenge were ascertained and compared with a commercial inactivated PCV2 commercial vaccine (Fostera^®^ Gold PCV; Zoetis Animal Health, Parsippany-Troy Hills, NJ, USA). Further, the CSFV-specific neutralizing antibody response in pigs vaccinated with PRVtmv+ was determined. Following the PCV2b challenge, viremia, virus load in the lymphoid tissues, leukocytes and lymphocyte counts, nasal- and fecal-virus shedding was compared between the unvaccinated and the prototype PRVtmv+ or the commercial (Zoetis), Fostera^®^ Gold PCV (Killed PCV type 1-type 2 chimera) vaccinated pigs. Data presented in this report documents that PRVtmv+ is highly attenuated and safe for vaccination in pigs. The PRVtmv+ yielded better protection for vaccinated pigs than the commercial vaccine after the PCV2b challenge regarding viremia, viral loads in the lymphoid tissues, and leukocyte or lymphocyte counts. Additionally, the PRVtmv+ vaccinated pigs generated low to moderate levels of CSFV-specific neutralizing antibody titers.

## 2. Materials and Methods

### 2.1. Cells and Medium

Swine kidney (SK; #CRL-2842, ATCC^®^, Manassas, VA, USA), Madin Darby bovine kidney (MDBK; #CCL-22, ATCC^®^), 293T (#CRL-3216, ATCC^®^), and TK negative TK-ts13 hamster cells (ATCC #1632) were propagated in Dulbecco’s modified Eagle’s medium (DMEM; #10-017-CV, Corning^®^, Corning, NY, USA) supplemented with 10% heat-inactivated fetal bovine serum (FBS; Equa1FETAL, Atlas Biologicals, Fort Collins, CO, USA) and 1 × antibiotic-antimycotic solution [34-004-CI, Corning^®^] (Growth medium). Plasmids and viral DNA co-transfection in 293T cells were performed using Opti-MEM^®^ (#31985-070; Gibco, Waltham, MA, USA) and lipofectamine 2000 (#11668030, Thermo Fisher Scientific^®^, Waltham, MA, USA).

### 2.2. Viruses

PRV wild type (wt) Becker strain is a virulent field isolate from a dog at Iowa State University, with subsequent laboratory passage [39] and low passage virus stock being propagated and maintained at −80 °C. Infectious PCV2b plasmid DNA clone [40] was obtained from Dr. Rowland (University of Illinois), and the virus was reconstituted by transfection into SK cell using polyethyleneimine (#23966, Polysciences, Warrington, PA, USA) as described previously [40]. Reconstituted PCV2b virus stock was titrated in SK cells. PRV wt and the genetically engineered recombinant PRV viruses were propagated in SK cells and titrated in MDBK cells, as described earlier [41]. Aliquots of low passage viral stocks were maintained at −80 °C. The CSFV strain Brescia (genotype 1.1) was obtained from the CSF virus collection of the EU and OIE Reference Laboratory for CSF (Institute of Virology, University of Veterinary Medicine, Foundation, Hannover, Germany) [42].

### 2.3. Antibodies

PRV-specific rabbit anti-gE antibody was kindly provided by Lynn W. Enquist, Princeton University, NJ, USA. CSFV E2 (HC/TC 50/2/1 and E^rns^ (HC/TC 169/2/3) specific mAbs were kindly provided by Paul Becher, University of Veterinary Medicine, Hannover, Germany. Anti-PCV2 Cap rabbit polyclonal antibody was generated commercially (Genscript, Piscataway, NJ, USA) against a PCV2 Cap-specific *E. coli*-expressed polypeptide (AMTYPRRRYRRNGIFDPYVNYSSRHTIPQPFSYHS-RYFTPKPVLDSTIDYFQPNNKRNQLWLRLQTSRNVDHVGLGTAFESKYDQDYNIRVT-MYVQFREFNLKDPPLNP). Anti-PCV2 capsid protein-specific mAb 36F1was kindly provided by Dr. Zoltan, CEVA. Mouse anti-V5 mAb (#R960-25, Thermo Fisher Scientific^®^, Waltham, MA, USA) and anti-flag rabbit antibody (#F7425, Sigma-Aldrich^®^, St. Louis, MO, USA) were purchased. CSFV NS3-specific mouse mAb antibody BVD/C16 from Institute for Virology, Hannover, Germany, was used. Secondary antibodies, donkey anti-mouse IgG Alexa fluor 488 (#A-21202, Thermo Fisher Scientific^®^), goat anti-mouse IgG horseradish peroxidase (HRP) (#32430, Thermo Fisher Scientific^®^), goat anti-pig IgG HRP (#ab102135, Abcam, Waltham, MA, USA), rabbit anti-mouse HRP (Agilent Technologies, Waldbronn, Germany), and FITC-rabbit anti-porcine IgG-H+L [17-9111, Invitrogen] were purchased. Goat anti-mouse HRP conjugated antibody (#32430) and donkey anti-rabbit HRP conjugate (#31458) were purchased from Thermo Fisher scientific.

### 2.4. Construction of PRVtmv+ Vector Virus Expressing the CSFV E2, E^rns^-GM-CSF, and PCV2b Cap Proteins

To generate the PRVtmv+ vaccine virus (Figure 1), the gE-, TK-, and gG-deleted triple mutant virus (PRVtmv) vector was constructed first.

#### 2.4.1. Construction of PRV gE-Deleted Virus (PRV gEΔ)

Initially, a PRV gEΔ was generated by homologous recombination of full-length PRV wt DNA and a gE deletion plasmid (pPRV gEΔ, Figure 1C). Briefly, the pPRV gEΔ plasmid was constructed by PCR amplification of a 1167 bp EcoRI-HindIII fragment (partial gI ORF and gE promoter sequence; GenBank accession #JX797219, nt 120,858 … 122,024) and 936 bp KpnI-BamHI fragment (gE-Us9 intergenic, Us9 ORF, and partial Us2 sequence; GenBank accession #JX797219, nt 123,846 … 124,781) using the PRV wt Becker DNA as a template and the corresponding PRV gE left-flanking (F1-R1) and right-flanking primer pairs (F2-R2), respectively (Table 1). Subsequently, the 1167 bp EcoRI/HindIII and the 936 bp KpnI-BamHI fragments were cloned sequentially into the corresponding sites of the plasmid pGEM3Z. In the pPRVgEΔ clone, the entire gE ORF was deleted, and two restriction sites HindIII-KpnI sites were incorporated in the gE deletion locus for future directional CSFV E2 chimeric gene insertion (Figure 2, see below). A PRV gEΔ recombinant virus was plaque purified and selected to construct a PRV gE/TK dual gene-deleted virus.

#### 2.4.2. Construction of PRV gE/TK Dual Gene-Deleted Virus

The TAATA boxes for the TK upstream U_L_24 and TK downstream U_L_22 genes are located within the TK ORF coding sequences (Figure 1B). These two genes are essential for virus replication [43,44]. Therefore, to construct a TK null/deletion plasmid (pTKΔ) without affecting the surrounding U_L_24 and U_L_22 promoter sequences, PRV nucleotide sequences (GenBank accession #JX797219) spanning nt 58,548–59,519 (primer pairs F3/R3) and nt 59,571–60,474 (primer pairs F4/R4) were amplified by PCR as EcoRI/KpnI and HindIII-NsiI fragments, respectively. The two fragments were cloned sequentially into the corresponding sites of plasmid pGEM7Z (Promega Corporation, Madison, MA, USA). In the resulting clone, plasmid pPRV TKΔ (Figure 1D), 51 nt within the TK ORF (nt 59,520 … 59,570), immediately downstream of the TK aa residue 135 were deleted, and instead the KpnI-HindIII sites were inserted. Additionally, immediately upstream of the KpnI site, three stop codons were incorporated (Figure 1D). Therefore, the coding sequences of TK aa residues 136–320 were truncated, and in the context of the viral genome, the TK gene is inactivated due to these deletion-insertional manipulations. The intended mutations within the TK coding region of the pPRV TKΔ DNA were verified by sequencing with a series of sequencing primers, F5–F8 and R5–R8 (Table 2). The two incorporated restriction sites, Kpn-HindIII in the TK ORF-deletion locus, allowed the chimeric PCV2b gene insertion (see below). One PRV gE/TK dual gene-deleted virus recombinant was selected to construct a PRV gE/TK/gG triple gene-deleted virus.

#### 2.4.3. Construction of PRV gE/TK/gG Triple Gene-Deleted (PRVtmv) Vector

The PRV gG ORF is located downstream of the U_S_3 ORF (Protein kinase; PK) and upstream of U_S_6 (gD). To generate a PRV gGΔ plasmid (pPRV gGΔ; Figure 1E), a 2041 bp long DNA fragment was synthesized and cloned into EcoRI (5′)/HindIII (3′) sites of pUC57 (BioMatik, Ontario, Canada). This fragment consisted from 5′ to 3′ of: an EcoRI restriction site, a 1001 bp PRV gG upstream flanking sequence comprising 5′–3′ direction, partial carboxy-terminal U_S_3 ORF, and Us3-Us4 intergenic sequences, (GenBank accession #JF797219; nt 116,964 … 117,964), a chimeric U_S_3 and U_S_4 polyadenylation (Poly A) signal as it is in the genome (GenBank accession #JF797219, nt 119,520 … 119,531) but placed immediately upstream of the Us4 start codon (ATG), followed by the KpnI restriction site (GGTACC), a 10 bp long non-genomic spacer sequence, the BamHI restriction site (Figure 1E) and the coding sequence for the PRV gG residues 72-, Us4 (gG)-gD (Us6) intergenic sequence (GenBank accession #JF797219; nt 118,177…119,176), and the restriction site for HindIII.

In the resulting plasmid pPRV gGΔ, nt 117,965 to 118,176 coding for gG residues 1 to 71 were deleted and replaced with the chimeric polyA and KpnI/BamHI sites, followed by the gG ORF residues 72–499 coding sequence and a HindIII restriction site. Consequently, in the context of the viral genome, the gG ORF sequence deletion would not affect the Us3 gene transcription. However, gG residues 72–499 aa will not be translated due to the insertion inactivation of the gG gene. In addition, the KpnI-BamHI sites could be utilized to insert the chimeric CSFV E^rns^-GM-CSF gene in a site-specific manner (see below).

### 2.5. Construction of PRV gEΔ CSFV-E2, PRV TKΔ PCV2 Cap and gGΔ CSFV Erns-GM-CSF Insertion Plasmids

#### 2.5.1. Construction of PRV gEΔ CSFV-E2 Insertion Plasmid (pPRVgEΔ/CSFV E2-INS)

To insert the chimeric CSFV E2 gene, for the construction of a PRV gEΔ CSFV E2 insertion plasmid (pPRV gEΔ/CSFV E2-INS), first, an ORF-less 1635 bp Pre CSFV E2 chimeric sequence was synthesized and cloned into KpnI (5′)/BamHI (3′) sites of pUC57 (Genscript), resulting in pPre CSFV E2 chimera (Genscript). The 2023 bp pPre CSFV E2 chimera sequence consisted of 5′–3′ direction as follows: a KpnI site, the 1662 bp nucleotide sequence for the CAG promoter (GenBank accession #GU299216.1, position 3–1664, which include CAG enhancer 3–364), plus a restriction site for NheI, a NcoI restriction site after a 12 bp spacer, 6× His coding sequence plus V5 epitope coding sequence (69 bp), followed by a stop codon, a bovine growth hormone (BGH) Poly A sequence (253 bp), and a BamHI restriction site. Second, a CSFV E2 chimeric ORF coding sequence was synthesized and cloned into pUC57 after codon optimization for pig (pCSFV E2 chimeric ORF; Genscript). The 1194 bp CSFV E2 chimeric ORF coding sequence consisted of 5′–3′ direction as follows: a NheI site, a Kozak sequence, PRV glycoprotein D signal sequence (gD predicted aa residues 1–18; GenBank accession #YP068387), predicted 373 aa of CSFV E2 ORF coding sequence (GenBank accession #AAC62087, aa 690 … 1062), and a NcoI site. Next, the 1194 bp NheI/NcoI fragment containing the chimeric CSFV ORF was cloned into the corresponding NheI/NcoI sites of pPre CSFV E2 chimera above. In the resulting clone (pCSFV E2 chimeric gene cassette) (Figure 2D), the expression of the CSFV E2 ORF with the PRV gD signal sequence is regulated by the CAG promoter and fused in frame with the V5 epitope and poly His coding sequences at the carboxy end. Lastly, the KpnI/BamHI fragment of pCSFV E2 chimeric gene cassette containing the CSFV E2 ORF (Figure 2D) was cloned into the corresponding KpnI/BamHI sites of pPRV gEΔ (Figure 1), resulting in pPRV gEΔ/CSFV E2-INS (Figure 2D, Appendix A).

#### 2.5.2. Construction of PRV TK-Deleted PCV2 Cap Insertion Plasmid (pPRV TKΔ/PCV2 Cap-INS)

To construct the pPRV TKΔ/PCV2 Cap insertion plasmid (pPRV TKΔ/PCV2 Cap-INS), an ORF-less 1638 bp pre-PCV2 Cap chimeric sequence (pPre PCV2 Cap) was synthesized (Genscript). The 1638 bp pre-PCV2 chimeric sequence consists of 5′–3′ direction as follows: a KpnI site, the nucleotide sequence for the human elongation factor 1α (hEF-1α) promoter (GenBank accession #J04617), a restriction site for NheI, a NotI restriction site after a 12 bp spacer, V5 epitope coding sequence, 6× His coding sequence, the simian virus 40 (SV40) Poly A sequence, and a HindIII restriction site. Second, after codon optimization for pigs, the chimeric PCV2 Cap ORF coding sequences were synthesized (pPCV2 Cap chimeric ORF; Genscript) along with a NheI site and Kozak sequence (at 5′ end) and a NotI site (at 3′ end). The 654 bp chimeric PCV2 Cap chimeric ORF coding sequence consists of 16 predicted amino acids of the PRV glycoprotein D signal sequence lacking the putative cleavage site (GenBank accession #YP068387; 1–16 aa) and a 654 bp NheI/NotI chimeric codon-optimized (for pig) sequence coding for 202 predicted amino acids of PCV2 Cap protein (aa 1–10 plus aa 42–233; GenBank accession # AAD45581) lacking its nuclear localization signal (residues 11–41; GenBank accession # AAD45581). Next, the 654 bp NheI/NotI fragment was cloned into the corresponding NheI/NotI sites of pPre PCV2 Cap chimera synthesized above. In the resulting clone (pPCV2 Cap chimeric gene cassette), the chimeric PCV2 Cap gene is regulated by the strong hEF-1α promoter and the PCV2 Cap ORF is fused in frame with the V5 epitope and poly His coding sequence at the carboxy end (Figure 2B). To assemble the pPRV TKΔ/PCV2 Cap-INS (Figure 2E, Appendix A), the 2289 bp KpnI/HindIII fragment containing the PCV2 Cap chimeric gene sequence (Figure 2E) was cloned into the corresponding KpnI/HindIII sites of pPRV TKΔ (Figure 2B).

#### 2.5.3. Construction of the gG Deletion/CSFV E^rns^-GM-CSF Insertion Plasmid (pPRV gGΔ/CSFV E^rns^-GM-CSF-INS)

To construct the pPRV gGΔ/CSFV E^rns^-GM-CSF-INS, a 2002 bp codon-optimized (for pig) KpnI (5′)/BamHI (3′) DNA fragment coding for the CSFV E^rns^-GM-CSF (porcine) chimeric ORF was synthesized and cloned into KpnI (5′)/BamHI (3′) sites of pBluescript (BioMatik). The 2002 bp KpnI/BamHI fragment consists of the following (5′ to 3′): a KpnI restriction site, a sequence for the cytomegalovirus (CMV) promoter sequence (GenBank accession #U55763; nt 1 … 605), a Kozak sequence, the PRV gD signal (GenBank accession #JF797219; nt 119,647 … 119,700, GenBank accession #YP068387; aa 1 … 18), the nucleotide sequence for CSFV E^rns^ (GenBank accession #AF091661; nt 1175 … 1855, GenBank accession #AAC62087; aa 268 … 494), followed by the nucleotide sequence of porcine GM-CSF (GenBank accession #AAM48280; aa 1, 18 … 144) fused in frame with the C-terminal E^rns^ coding sequence but lacking the stop codon, the nucleotide sequence for a flag tag (GACTACAAAGACGATGACGACAAG), a stop codon (TAA), the SV40 Poly A site (GenBank #U55763; nt 1411 … 1640), and the restriction site for BamHI. The 2.002 kb KpnI/BamHI fragment of pCSFV E^rns^-GM-CSF chimeric gene cassette (Figure 2F) was cloned into the KpnI/BamHI sites of pPRV gGΔ, resulting in pPRV gGΔ CSFV E^rns^-GM-CSF-INS (Figure 2F, Appendix A). Consequently, in the context of the viral genome, due to the inactivation of gG gene (deletion of the amino-terminal amino-terminal 71 aa coding sequence, and insertion of 24 nucleotides) (Figure 1) and the insertion of the chimeric CSFV E^rns^-GM-CSF (Figure 2), instead of gG, the chimeric CSFV E^rns^-GM-CSF would be expressed as a partially secreted protein. The nucleotide sequence of the pPRV gGΔ/CSFV E^rns^-GM-CSF-INS was verified and the expression of E^rns^-GM-CSF was verified by transfection of the plasmid DNA in the SK cells and immunoblotting with the CSFV E^rns^-specific mAbs.

#### 2.5.4. Construction of PRVtmv Vector Virus Expressing CSFV E2 and E^rns^-GM-CSF, and PCV2b Cap Chimeric Genes (PRVtmv+)

To construct the PRVtmv+, the PCV2b Cap chimeric gene was first incorporated in the PRVtmv viral genome by cotransfection and homologous recombination of PRVtmv genomic DNA and linearized pPRV TKΔ/PCV2 Cap-INS constructed above. PCR identified putative recombinant viral plaques were plaque purified and verified further by sequencing and immunoblotting with PCV2 Cap-specific rabbit polyclonal antibody. A selected PRVtmv expressing the PCV2b Cap was then chosen to sequentially incorporate the CSFV-E2 and E^rns^-GM-CSF chimeric genes by cotransfection of the PRVtmv-PCV2 Cap genomic DNA with the corresponding linearized insertion plasmids, pPRV gEΔ/CSFV E2-INS, and pPRV gGΔ/CSFV E^rns^-GM-CSF-INS, respectively. In each case, the putative recombinants were verified by PCR followed by sequencing and immunoblotting using the corresponding CSFV E^rns^- or E2-specific mAbs, respectively. One PRVtmv+ recombinant virus expressing all three subunit chimeric antigens was selected for further in vitro characterization.

### 2.6. Virus Titrations

PRV wt and the PRV recombinant viruses were titrated by plaque assay in MDBK cells, as described previously [41]. For PCV2, infected cells in 24-well titration plates were fixed with 3% paraformaldehyde (PFA; #30525-89-4; Acros Organics BVBA, Fair Lawn, NJ, USA) in phosphate-buffered saline (PBS; #P3813, Sigma-Aldrich^®^), and non-cytopathic viral plaques were visualized by indirect immunofluorescence assay (IFA) using anti-PCV2 capsid protein (Cap)-specific mAbs 36F1. Fluorescent antibody (FA) labeled PCV2 plaques were counted under an inverted fluorescent microscope (Olympus IX71, Shinjuku City, Tokyo, Japan).

### 2.7. Growth Kinetics and Plaque Size Assay

Growth kinetics of PRVtmv+ was evaluated and compared with that of the PRV wt by standard one-step growth kinetics assay, as described previously [41,45]. To determine the cell-to-cell spread property of PRVtmv+ compared with that of PRV wt, average plaque sizes of wt and PRVtmv+ viruses were determined by measuring approximately 50 randomly selected plaques of each virus group under a microscope with a graduated ocular objective, as previously described [46,47].

### 2.8. Animals and Experimental Design

Animal handling, sample collection, immunization, challenge infection, and euthanasia protocols were approved by the LSU Institutional Animal care and Use Committee (Protocol #20-027). Fifteen four week-old healthy Yorkshire pigs were purchased from a PCV2 free supplier (Valley Brook Research, Madison, GA, USA). Before inclusion in the study, pigs were tested for bovine viral diarrhea-free status by the serological assay as described previously [41]. After acclimatization for seven days, pigs were divided randomly into three groups of 5 pigs each. Group 1 (Control group; sham immunization), group 2 immunized with the commercial (Zoetis), Fostera^®^ Gold PCV (Killed PCV type 1-type 2 chimera) vaccine, and group 3 (PRVtmv+ vaccine group), vaccinated with the prototype vaccine. The pigs in groups 2 and 3 were housed in pens, at least 100 feet apart, in the pole barn-large animal isolation facility at the School of Veterinary Medicine, Louisiana State University. The pigs in the control group were housed in a separate swine barn, approximately 100 yards away from the pole barn. All sanitary precautions were taken to prevent cross-contamination between the groups. Footbaths were located at the entrance of the pole barn, and in front of each pen entrance. All bedding materials and excretions from pigs were sterilized before discarding.

#### 2.8.1. Vaccination and Challenge

The vaccination and PCV2b challenge schemes are shown in Figure 3. Each pig in the PRVtmv+ vaccine group was vaccinated intranasally (IN) with 4 × 10^7^ PFUs per nostril (total 8 × 10^7^ PFU) and subcutaneously (SC) with filtered (0.2 µm pore size) 4 × 10^7^ PFUs. According to the manufacturer’s instructions, the pigs in the Fostera vaccine group were vaccinated intramuscularly (IM) with 2 mL of the vaccine. The pigs in the control group were sham inoculated IN with 1.0 mL of cell culture media. The pigs in the PRVtmv+ vaccine group received Noromycin^®^ 300 LA (Norbrook, Lenexa, KS, USA) 20 mg/kg of body weight (BW) IM. At 32 days post-vaccination (dpv), animals of all three groups were challenged with PCV2b IN, with a total of 1.6 × 10^4^ PFU (8 × 10^3^ PFU/nostril) and SC with 6.75 × 10^3^ PFUs.

#### 2.8.2. Clinical Examination of Pigs Following Vaccination and Challenge

Pigs were routinely monitored for any visible clinical illness, feed, and water intake every day. Rectal temperature, BW, and clinical signs were recorded on 0, 2, 4, 5, 6, 8, 15, 21 dpv, and 32 dpv/0 days post-challenge (dpc). Following the PCV2b challenge, pigs were examined daily, and body temperatures and weights were recorded on 7, 13, 17, and 21 dpc. Clinical evaluation included rectal temperature, injection site reactions, depression, lethargy, sneezing, coughing, ocular-oro-nasal discharge, diarrhea, and systemic illness and lesions, if any.

#### 2.8.3. Sample Collection and Processing from the Vaccinated and Control Pigs following Vaccination and PCV2b Challenge

The scheme of sample collection (Ethylenediaminetetraacetic acid [EDTA]-blood, serum, nasal, fecal, and tonsil swabs) is shown in Figure 3. The swabs were collected in 2 mL of DMEM, supplemented with 3× antibiotic-antimycotic solution and 2% FBS. Collected swab samples were aliquoted and stored at −80 °C until use. Blood samples collected for sera were processed, aliquoted, and stored at −80 °C. Peripheral blood mononuclear cells (PBMCs) were separated from EDTA-blood using Ficoll-Paque^TM^ Plus (GE Healthcare, Chicago, IL, USA) and cryopreserved in liquid nitrogen, as described previously [41].

#### 2.8.4. Leukocyte and Lymphocyte Counting in Whole Blood

For counting the leukocytes and lymphocytes in whole blood-EDTA samples, an automatic hematological analyzer (Advia 120; Siemens Healthcare Diagnostics, Tarrytown, NY, USA) was used. On the day of vaccination (0 dpv), challenge (0 dpc), and 21 dpc, total leukocyte and lymphocyte counts were determined and recorded. In addition, the percent decline in leukocyte and lymphocyte numbers in each pig was calculated as follows and described earlier [41].
$$\begin{array}{c} \%\;\mathrm{change}\;\mathrm{in}\;\mathrm{leukocyte}\;\mathrm{count}\;\mathrm{of}\;\mathrm{pig}=100-\\ \frac{\mathrm{Leukocyte}\;\mathrm{count}\;\mathrm{of}\;\mathrm{the}\;\mathrm{animal}\;\mathrm{at}\;21\;\mathrm{dpc}}{\mathrm{Leukocyte}\;\mathrm{count}\;\mathrm{of}\;\mathrm{the}\;\mathrm{animal}\;\mathrm{on}\;0\;\mathrm{dpc}\;}\times 100 \end{array}$$

#### 2.8.5. Euthanasia, Necropsy, Tissue Sample Collection, and Processing

Pigs were euthanized with Euthasol^®^ (Euthanasia Solution; pentobarbital sodium and phenytoin sodium) and xylazine at 21 dpc. After a complete necropsy examination, tissue samples were collected from tonsils, lungs, liver, spleen, kidney, Peyer’s patches, and cervical, bronchial, mediastinal, and mesenteric lymph nodes (LN) for histopathological (10% formalin), virus isolation, and qPCR assays (dry ice). Formalin-fixed tissues were paraffinized, sectioned, and processed either for histopathology (H&E staining) or immunohistochemistry.

### 2.9. Serum Virus Neutralization (SN) by Plaque Reduction Assay for PRV and PCV2b

A standard plaque reduction assay was performed to evaluate the PRV- and PCV2b-specific virus-neutralizing antibody titers in serum, using 100 PFUs, as described previously [41]. The plaque reduction assay was performed as above with some modifications. Since the PCV2b is non-cytopathic, plaques were visualized by FA staining with a PCV2b Cap-specific mAbs and then counted under a fluorescent microscope as described earlier [41]. The virus-neutralizing antibody titers for each serum sample were estimated by calculating the highest dilutions of the serum that neutralized 50% of the average numbers of respective control virus plaques without serum.

### 2.10. CSFV-Specific SN Assay

CSFV-specific neutralization test for the PRVtmv+ vaccinated pigs sera was performed according to the protocol of the Manual of Diagnostic Tests for Detection of CSF, which was composed by the EU and OIE Reference Laboratory for CSF and is available on the website of the EU and OIE Reference Laboratory for CSF (https://www.tiho-hannover.de/kliniken-institute/institute/institut-fuer-virologie/eu-and-oie-reference-laboratory, accessed on 22 October 2021) [48]. The titration of the antisera started with a 1:2 dilution and was incubated with the CSFV strain Brescia (CSFV genotype 1.1). At 72 h post-infection, the cells were fixated by heat treatment for four hours at 80 °C. CSFV antigen detection in the cells was performed by immune-peroxidase staining as described in the Manual of Diagnostic Tests for Detection of CSF [48] using NS3-specific monoclonal mouse antibody BVD/C16 (dilution 1:50) and the conjugate rabbit anti-mouse horseradish peroxidase (dilution 1:200). Serum neutralization dose (ND50) titers were calculated as described previously [49].

### 2.11. DNA Isolation and Quantitative PCR (qPCR)

To quantify the PRVtmv+ and PCV2b genome copies following vaccination and challenge, respectively, from swabs, sera, PBMCs, and tissue samples, total DNA was isolated using the QIAamp^®^ DNA mini kit (#51306, Qiagen, Hilden, North Rhine-Westphalia, Germany). In addition to the nasal and tonsil swabs (PRVtmv+ and PCV2b), fecal swabs were collected (PCV2b). For DNA isolation, 25 mg tissue were homogenized using 2.8 mm ceramic beads (#15-340-154, Thermo Fisher Scientific^®^) in Precellys 24 homogenizer (#13112, Bertin Instruments, Rockville, MD, USA) and DNA was isolated as above. PRVtmv+ and PCV2b-specific genome copies were determined by TaqMan probe-based Real-time qPCR in ABI PRISM™ 7900HT Sequence Detection System (Applied Biosystems, Waltham, MA, USA), using major capsid protein (VP5) ORF coding (PRV wt) and Cap-specific (PCV2b) primer pairs (Table 3). Each time, the PCR reaction setup was run with six standards of known quantity (10^1^ to 10^6^ copies per reaction). PRV or PCV2b genome copies in the samples were compared with the generated standard curves. Viral genome copies were normalized to a standard curve generated with host-specific swine housekeeping gene, Glyceraldehyde 3-phosphate dehydrogenase (GAPDH; GenBank accession #AF017079.1). The assay was performed in duplicates, and results were expressed as PRV or PCV2b genome copies per million cells, with the given fact that each eukaryotic diploid cell of pig has two copies of the GAPDH gene.

### 2.12. Immunohistochemistry

Immunohistochemical analysis of paraffinized tissue sections was performed for comparing PCV2b antigen distribution in different groups of pigs using the anti-PCV2 Cap-specific mAb 36F1. Following standard deparaffinization and rehydration procedures in xylene and alcohol, respectively, endogenous peroxidase activity was quenched by incubating slides in 3% hydrogen peroxide in methanol for 1 h at RT. Antigen retrieval was performed by incubating tissue sections with proteinase K (20 µg/mL) in Tris-EDTA buffer (50 mM Tris Base, 1 mM EDTA, 0.5% Triton X-100, pH 8.0) for 30 min at 37 °C, after permeabilization with (0.2% Triton X-100 in TBS for 15 min and blocking with 4% nonfat dry milk (Blotting-grade blocker, #170-6404, Bio-Rad, Hercules, CA, USA)) in Tris-buffered saline (TBS; 20 mM Tris and 150 mM sodium chloride; pH 7.4) for 1 h at 37 °C. The slides containing the tissue sections were washed 3–4 times (TBS with 0.05% Tween-20). The slides were then incubated with anti-PCV2 mAb at 37 °C for 1 hr. Finally, slides were incubated with goat anti-mouse IgG peroxidase conjugate at 37 °C for 1 h, followed by Sigmafast™ 3,3′-Diaminobenzidine tablets (#D4293, Sigma-Aldrich^®^) in distilled water as per manufacturer’s instruction and counter-stained with 0.5% methyl green for 5 min. Sections were dehydrated, cleared, and mounted with a glass coverslip. Dark brown positive signals detected the presence of PCV2b antigen in tissue sections under the microscope.

### 2.13. Transmission Electron Microscopy (TEM)

For TEM, SK cells were grown on a 13 mm electron microscopic coverslip (#174950, Thermanox plastic coverslip, Ted Pella Inc., Redding, CA, USA) and infected with PRVtmv or PRVtmv+ at a multiplicity of infection (MOI) 5 or PCV2b (0.1 MOI). Similarly, uninfected SK cells were grown on coverslip as healthy control cells. After 12 and 18 h post-infection (PRV wt and PRVtmv+, respectively), or 72 h post-infection (PCV2b), cells were fixed with primary fixative (1.25% glutaraldehyde and 2% formaldehyde in 0.1 M Cacodylate buffer) for 1 h. Fixed cells were washed with 0.1 M Cacodylate buffer for 10 min/three times each and post-fixed with 1% osmium tetroxide for 1 h. Following washing with 0.1 M Cacodylate buffer with 5% sucrose (pH 7.4), the samples were stained with 0.5% uranyl acetate in 0.2 M sodium acetate buffer (pH 3.5) overnight. Samples were dehydrated in an ascending series of ethanol, infiltrated with epoxy resin to propylene oxide, embedded beam capsules, and polymerized at 60 °C overnight. Semi- and ultrathin sections were cut, mounted on Nickel-grids, examined with an electron microscope (JEM-1400 TEM; Louisiana State University, shared instrumentation facility), and captured digital images for analysis.

### 2.14. Histopathology

Formalin-fixed tissues were processed, embedded in paraffin, and 4 µm sections were stained with H&E following standard procedures at the Louisiana Animal Disease Diagnostic Laboratory (LADDL). A single veterinary pathologist, blinded to the treatment groups, evaluated slides for any lesion. The lymphoid organs (cervical and mesenteric LN, spleen, tonsil, and Peyer’s patches) were investigated for lymphoid hyperplasia, lymphoid depletion, granulomatous inflammation, multinucleated giant cells, and marginal zone cellularity (except Peyer’s patches). Tonsils were also investigated for crypt inflammation. Lungs were investigated for Bronchus-associated lymphoid tissue (BALT), peribronchiolar lymphocytes, bronchial inflammation, alveolar macrophages, alveolitis, and inflammation of the interlobular septa/pleura. The livers were investigated for portal inflammation, hepatocellular damage, and parenchymal inflammation. The kidneys were investigated for interstitial inflammation, tubular damage, glomerular damage, and pelvic inflammation.

### 2.15. Statistical Analysis

The sample size was calculated following a power analysis of the primary variable of interest to maximize confidence in that metric. Our data from a pilot study concerning serum virus neutralizing titers were used to estimate the expected differences (i.e., differences of 0.90 for viral titers) among our treatment(s) at each time point, 7, 14, and 21 days post-vaccination. As our data suggest, these days are most relevant for detecting differences in our variables of interest. Means and associated variances from our preliminary studies were utilized in our sample size calculation, with a desired power of 90%. This resulted in a sufficient size of 5 pigs per group.

Statistical analyses were performed using GraphPad PRISM^®^ software version 5.04 (GraphPad Software, San Diego, CA, USA). The two-way analysis of variance (ANOVA) followed by Bonferroni post-tests to compare replicate means by row were performed. A value of *p* ˂ 0.05 was considered statistically significant.

## 3. Results

### 3.1. Characterization of PRVtmv+

The genomic sequence of one selected PRVtmv+ spanning the gE-, TK-, and gG- deletion regions genes containing the CSFV E2, PCV2b Cap, and the chimeric CSFV E^rns^-GM-CSF fusion genes were verified. Nucleotide sequence results validated the results obtained during the single, dual, and triple deletion steps (Appendix A).

Additionally, the PRVtmv+ virus was tested for its inability to grow in the TK-negative, TK-ts13 hamster cells relative to its SK and MDBK cells growth. The lack of the gE-specific band in the PRVtmv+ was also verified by immunoblotting the infected cell lysates with a gE-specific antibody.

#### 3.1.1. Characterization of PRVtmv+ for the Expression of the CSFV E2, PCV2b Cap, and Chimeric CSFV E^rns^-GM-CSF

The PRVtmv+ virus was further analyzed for the expression of chimeric CSFV E2, CSFV E^rns^-GM-CSF (E^rns^+), and PCV2 Cap proteins by immunoblotting with the CSFV E2- and Erns-specific mAbs, and rabbit PCV2b-Cap-specific polyclonal antibodies. As depicted in Figure 4, the CSFV E2-specific mAbs detected an approximate 53–55 kDa (E2 monomer) and an approximate 103–110 kDa (putative dimer) band in PRV TMV+ -infected SK cell lysates (Figure 4). These two CSFV E2 chimeric protein bands were absent in the mock- and PRV Becker (wt)-infected SK cell lysates. The monoclonal antibody against CSFV E^rns^ detected several bands; 41 kDa, 58 kDa, 80 kDa, and 120–160 kDa bands in the PRVtmv+-infected cell lysates. Native E^rns^ expressed by CSFV form homo-dimers through a disulfide bond with approximately 100 kDa [31]. Most likely, the 41 kDa and 58 kDa bands are the unprocessed and processed monomers of E^rns^, respectively. The 80 kDa and 120–160 kDa bands are most likely the unprocessed and processed dimers of E^rns^+, respectively. The molecular weight of the chimeric PCV2 Cap V5 His protein was predicted to be 28.1 kDa. The rabbit anti-PCV2 Cap antibody detected an approximate 56 kDa and 110 kDa bands in PRVtmv+-infected cell lysates. Therefore, the 55 kDa and 110 kDa bands recognized by the Cap-specific antibody are most likely the dimeric and tetrameric forms of the chimeric PCV2b Cap, respectively. The chimeric E2, E^rns^, and Cap proteins were absent in the case of mock- and PRV Becker (wt)-infected cell lysates.

#### 3.1.2. PRVtmv+ Expressed PCV2b Cap Self-Assembles into Virus-like Particles (VLPs) In Vitro

A transmission electron microscopy study was performed for morphological evaluation of PCV2b, PRV wt, and PRVtmv+ infected SK cells. Specifically, the emphasis was to determine whether the chimeric PCV2 Cap expressed in the PRVtmv+-infected cells can form PCV2b VLPs that resemble the PCV2b virus particles in the PCV2b-infected SK cells. Mock-infected healthy SK cells showed normal cellular morphology (Figure 5A,B). PCV2b infected cells at 72 hpi showed accumulation of PCV2 viral particles within the vesicle-like structures in the cytoplasm (Figure 5C,D). Each PCV2 particle measured about 20 nm in diameter. PRV wt virus-infected cells showed typical enveloped herpesvirus particles (about 200 nm in diameter) within the vesicular structures in the cytoplasm (Figure 5E–G). The PRVtmv+ vaccine virus-infected SK cells also showed enveloped PRVtmv+ virus particles (about 200 nm in diameter) within the vesicular structures in the cytoplasm at 18 hpi (Figure 6). Additionally, PCV2b VLPs within vesicular structures in the cytoplasm were detected, which resembled the PCV2b virus particles in the PCV2b-infected cells. The PCV2b-VLPs were circular, and each measured about 20 nm in diameter with an inter-particle distance of 5 nm. Therefore, the electron microscopic results demonstrated that the PCV2b capsid protein expressed in PRVtmv+ vector self-assembles into PCV2b VLPs.

### 3.2. PRVtmv+ Vaccine Virus Replicates with a Similar Kinetics and Virus Yield in SK Cells, In Vitro, Like the PRV wt but Produces Smaller Plaques

Two independent assays were performed to determine the plaque sizes and one-step growth kinetics of the PRVtmv+ virus relative to the PRV wt virus. As shown in Figure 7, the PRVtmv+ produced significantly smaller plaques (about 72% reduction in plaque size) compared to the PRV wt virus (Figure 7A,B). However, the one-step growth kinetics and virus yield of the PRVtmv+ virus was similar to the PRV wt. (Figure 7C; Appendix A).

### 3.3. PRVtmv+ Vaccine Virus Is Highly Attenuated, Safe, and Retains Its Stability in Pigs to Express the PCV2b and CSFV Chimeric Genes

All pigs were healthy and negative for PRV and PCV2b at the time of immunization. Two groups of five pigs were immunized with Fostera Gold PCV (killed vaccine) or live PRVtmv+ (PCV2b/CSFV subunit vaccine). Regardless of the vaccine used, all pigs were normal clinically after vaccination. Body temperature in immunized pigs was within the physiological range (38.67–39.78 °C), except that three pigs in the Fostera group had slightly higher body temperature (40.0–40.2 °C) at 6 dpv (Figure 8A; Appendix A). In the Fostera group, pigs did not gain weight until 6 dpv (Figure 8B; Appendix A). However, by 6 dpv, pigs in the control and PRVtmv+ vaccine groups had 15% and 3% weight gain, respectively, without statistical significance (*p* > 0.05).

### 3.4. Nasal Virus Shedding following IN/Subcut Vaccination with PRVtmv+ Vaccine

In PRVtmv+ immunized pigs, the vaccine virus replicated and shed in the upper respiratory tract as evidenced by viral plaque assay and PRV-specific qPCR (Figure 9 and Figure 10; Appendix A). On 2 and 4 dpv, most of the pigs showed PRVtmv+ nasal shedding by qPCR, but on day 8, only two pigs had virus nasal shedding (Figure 9A). Further, PRVtmv+ could be isolated in cell culture from two pigs’ nasal swab samples on 4 dpv (Figure 9B).

After natural infection, PRV replicates in the nasal, pharyngeal, and tonsillar epithelium [50] and spreads to other tissues. Therefore, we also determined the PRVtmv+ virus replication in the tonsil. The qPCR results shown in Figure 10 demonstrated that on 2, 4, and 8 dpv, all the pigs in the PRVtmv+ vaccine group had the vaccine virus in their tonsil. However, the vaccine virus was isolated from four pigs on 2 dpv and from only one pig on 4 dpv (Figure 10A,B; Appendix A). As expected, both control and Fostera group pigs remained negative for PRVtmv+ by qPCR and virus isolation, indicating no cross-contamination.

### 3.5. The Chimeric E2 and E^rns^ Proteins Are Intact in the PRVtmv+ Virus Isolated from the Vaccinated Animals at 4 dpv

To determine the stability of the PRVtmv+ vaccine virus after a passage in pig, in vivo, the virus isolated from nasal swabs on 4 dpv was tested by immunoblotting with anti-CSFV E2 and E^rns^-specific mAbs. As depicted in Figure 11, the E2- and E^rns^-specific mAbs recognized the 53 kDa E2- and 56 kDa E^rns^-specific bands, respectively.

### 3.6. PRVtmv+ Vaccinated Pigs Induced PC2b-Specific Antibodies in the Vaccinated Pigs, Which Detected PCV2b-Infected Cells in Culture

To test whether serum samples from the PRVtmv+ vaccinated pigs detected PCV2b-specific viral antigens in the PCV2b-infected SK cells, we tested sera samples from two vaccinated pigs collected at 0 dpv (before vaccination) and 32 dpv by immunofluorescence assay (IFA). As demonstrated in Figure 12, while the serum sample of a pig (#2313) collected before the vaccination did not react with the PCV2b antigens in the PCV2b-transfected cells, two pig (#2313 and 2304) sera samples collected on 32 dpv labeled the PCV2b non-cytopathic plaques in the similarly transfected cells, as visualized by fluorescent-tagged anti-pig antibodies (Figure 12). Therefore, the PRVtmv+ vaccinated pigs generated PCV2b Cap-specific antibodies.

Taken together, the results in Section 3.5 and Section 3.6 validated the stability of the vaccine virus and the expression of all three chimeric proteins in vivo.

### 3.7. PRVtmv+ Vaccine Stimulated a Reasonable PRV-Specific Neutralizing Antibody Response in the Vaccinated Pigs

At the time of immunization (0 dpv), none of the pigs among all three groups had any detectable levels of PRV- or PCV2b-specific SN antibody titers in the serum. However, by 8 dpv, detectable levels of PRV-specific neutralizing antibodies (mean SN titer of 10) were detected (Figure 13A; Appendix A). The PRV-antibody levels rose four-fold to 40 on 15 dpv and maintained slightly lower SN titers until 53 dpv (24 at 42 dpv and dropped to 11 at 52 dpv). None of the pigs in the control and Fostera group developed PRV-specific antibodies in serum throughout the study.

### 3.8. A Single Dose of PRVtmv+ Vaccine Is Sufficient to Elicit a Higher PCV2b-Specific Antibody Response Than the Inactivated Fostera Gold PCV Vaccine

Both PRVtmv+ and Fostera vaccine-induced detectable levels of PCV2b-specific neutralizing antibodies in the vaccinated pigs by 15 dpv (Figure 13B; Appendix A). At this point, the antibody level in the PRVtmv+ vaccine group was nearly two-fold higher than the Fostera groups’ titer; mean SN titer of 15 for the PRVtmv+ versus 8 for the Fostera group. The average SN antibody titers in the PRVtmv+ rose to 22 on the day of PCV2b challenge (32 dpv), while the corresponding average SN titer in the Fostera group was 16.

### 3.9. PRVtmv+ Immunized Pigs Generated CSFV-Specific Neutralizing Antibody Titers

To determine whether the PRVtmv+ vaccinated pigs also induced a CSFV-specific neutralizing antibody response, the ND_50_ titers against the CSFV strain Brescia was determined for sera samples collected on 0, 15, 21, and 32 dpv. On the day of vaccination (0 dpv), the pigs in the PRVtmv+ group had average ND_50_ titers of less than 2, but the average titers rose to 5, 8, and 11 on 15, 21, and 32 dpv, respectively (Figure 13C; Appendix A). Therefore, the pigs seroconverted (a four-fold rise) ND_50_ titers from less than 2 to 8 by 21 dpv and rose further to 11 on 32 dpv. These results demonstrate that the chimeric E^rns^ and E2 proteins expressed by the PRVtmv+ induced a low to moderate CSFV strain Brescia-specific serum neutralizing antibody titers.

### 3.10. Both the PRVtmv+ and the Fostera Vaccinated Pigs Had Similar PCV2b-Specific seroconversion after the Challenge, but the SN Antibody Titers in the PRVtmv+ Vaccine Group Were Two-Fold Higher than the Commercial Vaccine Group

Following the PCV2b challenge, the neutralizing antibody titers in both PRVtmv+ and Fostera groups continued to decline gradually through 13 dpc, from average SN titers of 23 (0 dpc) to 9 (PRVtmv+) and from average SN titers of 16 (0 dpc) to 6 (Fostera) (Figure 13B; Appendix A). However, by 21 dpc (day of euthanasia), pigs in both the groups SN titers increased more than a four-fold (seroconverted) 9 to 44 (PRVtmv+) and 6 to 28 (Fostera). Therefore, both vaccine groups had a similar delay in the seroconversion after challenge, but the average SN titers in the PRVtmv+ group were nearly two-fold higher than in the Fostera vaccine group (44 versus 28) (Figure 13B). Notably, antibodies against PCV2b in the control group begin to appear in a few pigs (two pigs out of five) on 13 dpc and rose to 13 on 21 dpc (Figure 13B).

### 3.11. While PRVtmv+ Vaccinated Pigs Had a Moderate Increase in Leukocyte and Lymphocyte Counts following the PCV2b Challenge, Pigs in the Control Unvaccinated and Commercial, Fostera Vaccine Groups Had a Reduction in Both the Counts (a Moderate to Low Level of Leukopenia and Lymphopenia)

PCV2b-infected piglets, especially when developing PCV2b associated PMWS disease, have lymphopenia [52,53]. Therefore, we determined the leukocyte and lymphocyte counts in control unvaccinated, Fostera and PRVtmv+ vaccinated pigs before challenge (0 dpc) and after the PCV2 challenge (at 21 dpc). The data presented in Figure 14A,B (Appendix A) showed that in the PRVtmv+ vaccinated pigs, both the leukocyte and lymphocyte counts increased by 21 dpc relative to 0 dpc (36% and 20%, respectively). At the same time, based on the criteria described in the Methods section for the leukopenia and lymphopenia, all five pigs in the control group developed a low to moderate level of leukopenia and lymphocytopenia averaging an approximate 29% decline (Figure 14A,B). Notably, four out of five pigs in the Fostera group also developed mild leukopenia and lymphocytopenia ranging between 4–36%, with an average decline of 20%. The remaining one pig in the Fostera group had a 36% increase in leukocyte count.

### 3.12. Both PRVtmv+ and Fostera Immunized Pigs Did Not Shed the PCV2b Challenge Virus in Feces

Fecal virus shedding is a transmission source of PCV2b virus in pigs [54]. We determined PCV2b shedding by qPCR in fecal swabs on 0, 13, 17, and 21 dpc. Low fecal PCV2b shedding levels were observed only in a few control group pigs on 17 and 21 dpc; PCV2 genome copy numbers ranged between 13–35 copies/200 ng of total DNA in two pigs on 17 dpc and one pig on 21 dpc (Figure 15; Appendix A). No PCV2b genomic DNA was detected in the fecal swabs of PRVtmv+ and Fostera-vaccinated groups after the PCV2b challenge through the day of euthanasia (21 dpc). Regardless of the experimental group, neither PCV2b infectious virus nor genomic DNA was detectable in the nasal and tonsil swabs after the challenge.

### 3.13. PRVtmv+ Protected the Vaccinated Pigs from Both Cell-Free and Cell-Associated Viremia after PCV2b Challenge

PCV2b infection causes both cell-free and cell-associated (monocytes and lymphocytes) viremia following replication in monocytes/macrophages and lymphoblasts in the lymphoid tissues [55]. Therefore, we tested whether the PRVtmv+ vaccination protected the pigs both from cell-free or cell-associated viremia after a PCV2b challenge. As depicted in Figure 16A, after the PCV2b challenge, PCV2b genomic DNA could not be detected by qPCR in the sera of pigs vaccinated with PRVtmv+ on 13, 17, and 21 dpc. However, PCV2 genome copies were readily detected in the sera of three pigs (13 dpc) and two pigs (17 and 21 dpc) of the control unvaccinated group; mean copies 116 on 13 dpc, 73 on 17 dpc, and 18 on 21 dpc (Figure 16A; Appendix A). In the case of Fostera group, PCV2b DNA was detected in sera of two pigs on both 13 dpc (44 and 38 copies, respectively) and 21 dpc (19 and 12 genomic copies) (Figure 16A). Therefore, the PRVtmv+ vaccinated pigs were entirely protected from cell-free viremia, whereas pigs receiving the commercial Fostera group had significantly reduced viremia compared with the control unvaccinated pigs.

To determine whether pigs in the PRVtmv+ vaccine group were also protected entirely from cell-associated viremia relative to the commercial vaccine group, PBMCs collected on 0, 7, 13, and 21 dpc from the vaccinated and unvaccinated control groups were analyzed by PCV2b-specific qPCR as above. The results depicted in Figure 16B (Appendix A) show that, while the control pigs had a high number of PCV2 genomic copies starting on 7 dpc until 21 dpc, when the pigs were euthanized, the pigs in the Fostera group had moderate numbers of PCV2b genomic copies only on the 13 dpc. Notably, the PRVtmv+ vaccinated group was again negative for PCV2b-specific DNA the entire time post-PCV2b challenge. Therefore, taking together the cell-free and cell-associated viremia data, the PRVtmv+ vaccine prevented viremia entirely. However, the commercial vaccine “Fostera” did not; the viremia was reduced from the high level in the control animals to a low to moderate level.

### 3.14. Gross and Histopathology Lesions in Pigs

Overall, pigs from the three treatment groups had no gross or histologic changes, with no evidence of lymphoid depletion and/or granulomatous inflammation within lymphoid tissues (Figure 17). In one pig vaccinated with the PRVtmv+ (#2302), rare multinucleated giant cells were noted in the mesenteric lymph node (Figure 17F); however, there was no lymphoid depletion or histiocytic inflammation, PCV2 DNA, identifiable viral botryoid inclusion bodies, or viral antigen. This is not surprising, as PCV2 usually requires a coinfection to produce lesions.

### 3.15. PRVtmv+ Vaccine Protected Pigs against PCV2b Challenge Better than the Inactivated Commercial Vaccine Fostera Based on Viral Load in Lymph Nodes

For viral load in the tissues, tissues were cut into several small pieces and snap-frozen in dry ice. Total DNA isolated from tissues was analyzed by qPCR to determine the viral load in the tissues. The PCV2b genome copy numbers were calculated after normalization with the porcine house-keeping gene GAPDH. The results depicted in Figure 18 (Appendix A) show that all pigs in the control group had the PCV2b genome in both tonsil and cervical LN with average copy numbers of 326 and 802 per one million cells, respectively. Three out of five control pigs had PCV2b in Peyer’s patches and mesenteric LN (mean genome copy numbers of 5257 and 263, respectively). Finally, two out of the five control pigs had PCV2b in mediastinal LN and spleen (mean genomic copy numbers 55 and 3000 copies, respectively. In contrast, no PCV2b viral genome copies were detected in any tissues of pigs immunized with PRVtmv+, except #2302, which had very low levels of PCV2b only in the mediastinal LN (22 genome copies). In the Fostera group, two out of five pigs had PCV2b genomes in mediastinal and cervical LN (mean genome copy numbers 48 and 14, respectively), and one pig (#2305) had 23 genomic copies in the tonsil and another pig (2310) had 861 copies in the Peyer’s patches. Taken together, PRVtmv+ protected the pigs better than the Fostera vaccine for PCV2b load in the lymphoid tissues (*p* ˂ 0.05).

### 3.16. With the Exception of One Pig in the Fostera Group, Both PRVtmv+ and Fostera Vaccinated Pigs Were Negative for PCV2b Cap-Specific Antigens in the Lymphoid Tissues

Immunohistochemistry was performed to demonstrate the PCV2b antigen in pig tissues. The results in Figure 19A,B, show that while PCV2b Cap antigen was detected in the tonsil and Peyer’s patches of pigs in the control unvaccinated pigs, none of the PRVtmv+ vaccinated pigs had detectable PCV2b Cap antigens in any tissues. In the Fostera vaccine group, only one pig (#2310) had detectable PCV2b Cap antigens in the Peyer’s Patches. As noted above, this pig also had 861 copies of PCV2b genome copies in the Peyer’s patch.

## 4. Discussion

This study constructed PRVtmv, a triple mutant virus lacking the entire gE gene, a serological marker distinguishing the vaccinated from the wt virus-infected pigs (DIVA). Additionally, the TK and gG genes were inactivated. Subsequently, chimeric CSFV E2 and E^rns^-GM-CSF genes were inserted in the gE- and gG- deletion loci, respectively. Furthermore, a chimeric PCV2b Cap gene was inserted in the TK deletion locus, resulting in a live-attenuated genetically engineered trivalent PRVtmv+ vaccine against PRV, PCV2b, and CSFV. We determined that the PRVtmv+ is safe for pigs and that it elicited virus-neutralizing antibodies in the vaccinated pigs against all three viruses; PRV, PCV2b, and CSFV. Notably, compared with the commercial Fostera Gold PCV vaccine, the PRVtmv+ induced approximately 2-fold higher PCV2b-specific neutralizing antibody titers after vaccination and after the PCV2b challenge. Both PRVtmv+ and Fostera prevented fecal virus-shedding of the challenge virus. However, only PRVtmv+ vaccination completely prevented pigs from both cell-free and cell-associated viremia as compared with the Fostera vaccine. Additionally, one PRVtmv+ vaccinated pigs had minimal PCV2b viral load only in the mediastinal lymph node. In contrast, several Fostera vaccinated pigs had low levels of viral load in the mediastinal and cervical lymph nodes. Notably, one Fostera pig had high PCV2b genome copy numbers in the Peyer’s patches.

Interestingly, similar to the control group pigs, most of the Fostera group pigs had leukopenia and lymphopenia. On the contrary, the PRVtmv+ vaccinated pigs had lymphocytosis and leukocytosis, indicating a better lymphoproliferative immune-stimulation by the chimeric PCV2b Cap antigen. Taking together, PRVtmv+ protected the pigs better than the Fostera by inducing a superior protective immune response. Notably, the results also demonstrated that the chimeric PCV2b Cap protein, expressed in the PRVtmv+ infected SK cells, self-assembled to PCV2b VLPs. These results indicated that the chimeric Cap protein is intact functionally and antigenically. Additionally, VLPs are potent immunostimulatory molecules, which present highly dense repetitive epitopes on the surface a more authentic confirmation so that the immune system can easily recognize them [56]. Therefore, we believe that live PRVtmv+ expressing PCV2b also produced VLPs in vivo and induced a better protective immune response compared with the commercial Fostera vaccine, which is based on inactivated PCV1 with PCV2 Cap. While the inactivation method is not included in the data sheet of Fostera vaccine, it is reasonable to assume that the inactivation procedure most likely denatured the Cap protein resulting in altered conformation. Consequently, the Fostera PCV2 vaccine generated an inadequate protective immune response compared with that of PRVtmv+.

Most importantly, both the control and Fostera vaccinated pigs had leukopenia and lymphopenia, while the PRVtmv+ vaccinated pigs had leukocytosis. These findings are consistent with the cell-free and PBMC-associated viremia in the case of Fostera-vaccinated and control pigs upon challenge but not in the case of PRVtmv+ vaccinated pigs. PCV2 infection induces apoptosis in vitro and in vivo [57,58,59,60,61]. Apparently, in pigs, PCV2 infection can produce lymphopenia by inducing apoptosis of B lymphocytes [62]. Macrophage apoptosis also can be detected in the spleen of PCV2 infected mice [63]. Most likely, PCV2 infection and replication in the lymphocytes and monocytes of both the control and Fostera pigs induced apoptosis, resulting in leukopenia and lymphopenia.

For vaccination with PRVtmv+, we used both intranasal and subcutaneous routes. Our results demonstrated that PRVtmv+ replicated well in the upper respiratory tract epithelium and tonsil following infection. Subcutaneous vaccine administration also improves stimulation of immune cells, including dendritic cells. Per the manufacturer’s recommendations, the inactivated Fostera vaccine was given via intramuscular injection. GM-CSF is an attractive adjuvant for live vectored vaccines because of its ability to recruit antigen-presenting cells to the site of antigen synthesis and its ability to stimulate the maturation of dendritic cells [64]. We incorporated GM-CSF with the E^rns^ and expressed it as a fusion protein. Our results clearly showed that PRVtmv+ induced better neutralizing antibody response against PCV2b and completely protected against PCV2b viremia and lymphopenia. In contrast, there was leukocytosis, and lymphocytosis in the case of PRVtmv+ vaccinated pigs. We believe that collectively, the experimental vaccine produced a better immune response in the case of PRVtmv+ relative to that of Fostera.

Although the protective efficacy of PRVtmv+ against CSFV has yet to be tested, the results in this study showed that PRVtmv+ vaccinated pigs induced low to moderate levels of CSFV-specific neutralizing antibody titers. Recently, a similar mutant bovine herpesvirus type 1 (BoHV-1) based live subunit vaccine virus against BVDV type 2 was more efficacious and protective than the commercial trivalent MLV BoHV-1, BVDV types 1 and 2. The CSFV-specific neutralizing antibody titer was similar to the live BoHV-1 vectored BVDV subunit vaccine-induced BVDV-specific neutralizing antibody titers. Additionally, the memory neutralizing antibody response following a live BVDV-2 challenge was rapid and significantly higher than the commercial trivalent vaccine [41]. Based on these findings with BVDV-2 vaccine, the PRVtmv+ vaccinated pigs are expected to have similar rapid, memory neutralizing antibody responses and be protected against CSFV. PRVtmv+ vaccination and CSFV challenge experiment will be performed in the near future at the Plum Island Foreign Animal Disease Center, Orient, NY, USA.

In China, CSFV C-strain and PRV Bartha-K61 strain-based MLV vaccines have been used for nearly two decades to control CSF and pseudorabies, respectively. Initially, both vaccines were effective against their respective wild-type virus infection. However, CSFV became endemic in China and caused intermittent CSF outbreaks over time. The CSFV C-strain cannot be distinguished from the circulating endemic strain, and the vaccine strain may persist in the vaccinated animals; therefore, the use of MLV vaccine is problematic and might be contributing to the endemic CSFV problem. Additionally, a highly neurovirulent PRV strain was recently isolated from pigs vaccinated with PRV Bartha K-61 [15,16]. Notably, the Bartha K-61 vaccine does not protect the vaccinated pigs clinically against the neurovirulent PRV strain [15,16,17,18,65,66].

In many East European and Asian countries, including China, the coinfection rates of PRV with PRRSV, PCV2, and CSFV are high in intensive pig farms. In China alone, the rate of PRV coinfected with PRRSV was 36.0%, followed by 12.9% with PCV2 and 1.8% with CSFV, respectively [17]. Therefore, the emergence of highly neurovirulent PRV strain in Bartha K61vaccinated pigs could have been due to complex epidemiological interactions between the circulating PRV and co-infecting endemic PCV2, PRRSV, and/or CSFV virus(es).

Even though CSFV and PRV have been eradicated in the domestic pigs of North America and, with one exception in the EU (Romania), PRV is circulating in the wild boar populations in Europe and US. CSFV outbreaks in Germany in the 1990s and subsequently in the Netherlands, Italy, and Spain were due to indirect or direct contact with wild boar infected with CSFV or swill feeding [7,67,68]. Generally, the circulation of CSFV in wild boars of the European, Central American, and Caribbean countries will always remain a potential threat [11,19,69]. However, routine vaccination against CSFV is not allowed in the EU. Instead, strict surveillance of CSFV in wild boars and domestic pigs is being implemented.

Recently, the European Medicines Agency (EMA) obtained a license for the bovine viral diarrhea virus (BVDV), a cattle pestivirus, chimera, designated as “CP7_E2alf” (Suvaxyn CSF Marker, Zoetis Belgium SA, Belgium) in which the E2 encoding region of an infectious cDNA clone of the cytopathogenic BVDV strain “CP7” [70] was replaced with the E2 encoding sequences of CSFV strain “Alfort/187” [71]. Therefore, the immune response induced by the CP7_E2alf chimera-based vaccine is distinguishable from the immune response after infection with the wt CSFV. The “CP7_E2alf” vaccine is safe to use in calves under experimental conditions [72]. However, its safety under field conditions has not been evaluated to date. Currently, the CP7_E2alf vaccine is for emergency use only, such as during an outbreak of CSF in swine.

Pestiviruses have higher mutation rates both in vitro and in vivo. Therefore, vaccine production requires strict quality control protocol with respect to its genomic sequence integrity. Most importantly, the production facility for CSFV requires a highly contained environment because of its biosecurity concerns. PCV2 is difficult to grow in cell cultures, and the highest virus titers in cell culture are usually 5–8 × 10^3^ PFUs/mL. PCV2 also has the highest mutation rates among the DNA viruses [22]. Therefore, only subunit protein or inactivated vaccines are currently in use and are not adequately protective. These problems would not be encountered in the case of PRVtmv+ live vaccine virus.

Our data also demonstrated that the PRVtmv+ virus retains genomic stability for the PCV2b and CSFV chimeric genes after multiple cell culture passages and a single animal passage. Our data also revealed that (Pavulraj and Chowdhury, unpublished data, manuscript in preparation) the PRVtmv+ does not replicate in TG neurons upon dexamethasone-induced latency-reactivation, and there is no nasal virus shedding. Therefore, there is no risk of vaccine virus transmission and circulation in the pig population. The PRVtmv+ virus is also easier to produce in vitro because it replicates significantly better in cell cultures, with a titer ranging from approximately 5 × 10^7^–1 × 10^8^ PFUs/mL. In conclusion, the PRVtmv+ is a better replacement for the currently available inactivated PCV2 vaccine in the field in terms of vaccine production and efficacy.

## Figures and Tables

**Figure 1 vaccines-10-00305-f001:**
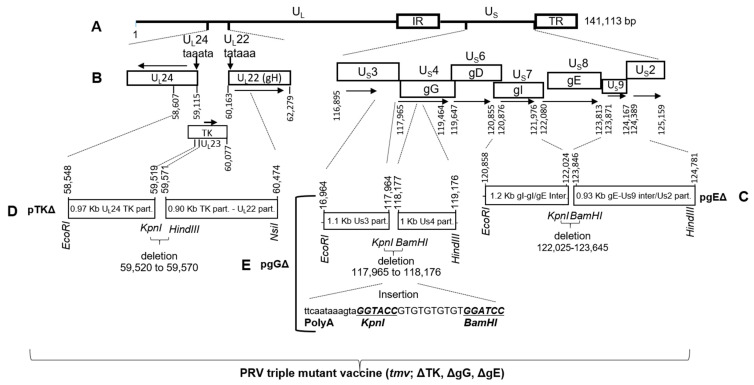
Schematic showing genomic configuration of pseudorabies virus (PRV) and the strategy of thymidine kinase (TK), glycoprotein G (gG), and gE gene deletion to generate a PRV triple mutant virus (tmv) vaccine vector. (**A**) Schematic of PRV wild type (wt) Becker strain backbone [39]. U_L_—Unique long region; U_S_—Unique short region; I_R_—Internal repeat region; T_R_—Terminal repeat region. Plasmid construct pPRV gEΔ (**C**) was used to introduce gEΔ in the PRV wt backbone (**A**) in the gE locus to generate PRV gEΔ mutant virus. Plasmid construct pPRV TKΔ (**D**) was used to incorporate TK deletion in the PRV gEΔ mutant backbone virus in TK locus (**B**) to generate PRV gE/TK dual gene deleted mutant virus. Finally, PRVtmv (gE/TK/gG-deleted) was constructed by incorporating the plasmid construct pPRV gGΔ (**E**) in to the gG locus in PRV gE/TK dual gene deleted mutant virus backbone. Arrows indicate the direction of the corresponding open reading frame (ORF). Nucleotide numbers correspond to the GenBank accession # JF797219. Part.—partial sequence; Inter.—intergenic sequence.

**Figure 2 vaccines-10-00305-f002:**
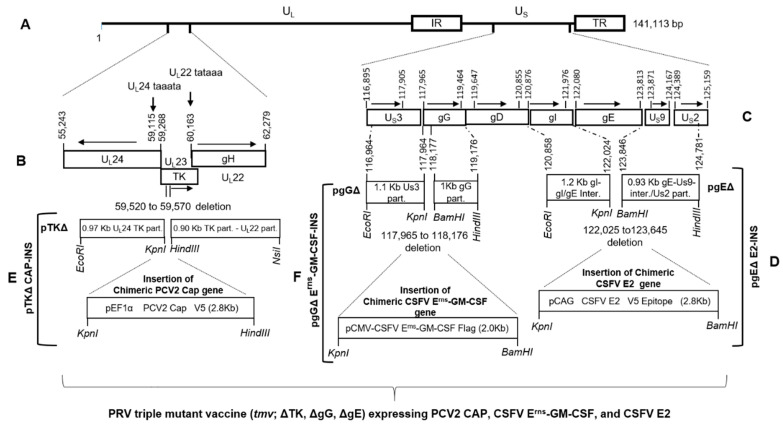
Strategy of chimeric PCV2 cap, CSFV E^rns^-GM-CSF, and CSFV E2 insertion in the TK deletion, gG deletion, and gE deletion loci, respectively, of PRVtmv genome to generate PRVtmv-CSFV E2-E^rns^-GM-CSF-PCV2b Cap (PRVtmv+). (**A**) Schematic of PRV triple mutant virus (tmv) genomic organization showing thymidine kinase (TK) (**B**), glycoprotein G (gG), and gE (**C**) deletions. Arrows indicate the direction of the corresponding open reading frame (ORF). Nucleotide numbers correspond the GenBank accession # JF797219. (**E**) Schematic of chimeric PCV2 cap V5 epitope gene expression cassette. The PCV2 cap expression cassette cloned in to EcoRI-NsiI site of pPRV TKΔ to yield pPRV TKΔ/PCV2 Cap-INS. (**F**) Schematic of chimeric CSFV E^rns^-GM-CSF Flag expression cassette. The CSFV E^rns^-GM-CSF expression cassette cloned in to KpnI-BamHI site of pPRV gGΔ to yield pPRV gGΔ/CSFV E^rns^-GM-CSF-INS (**D**) Schematic of chimeric CSFV E2 V5 epitope gene expression cassette. The CSFV E2 expression cassette was cloned in to KpnI-BamHI site of pPRV gEΔ to yield pPRVgEΔ/CSFV E2-INS. Part.—partial sequence; Inter.—intergenic sequence.

**Figure 3 vaccines-10-00305-f003:**
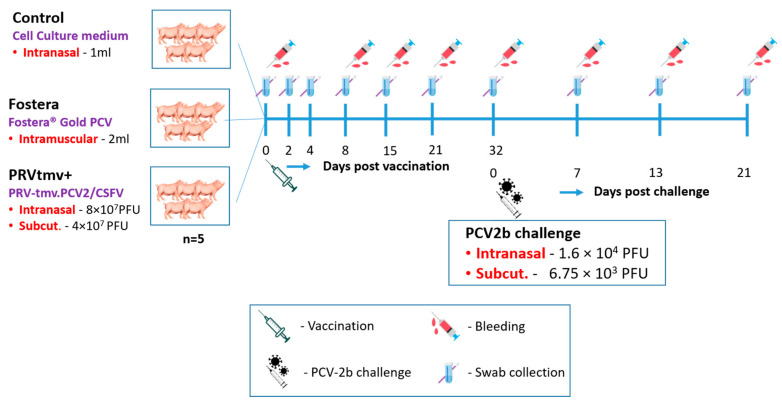
Vaccination, sample collection, challenge, and euthanasia scheme for the animal experiment. Intranasal (IN)—intranasal inoculation; Subcut—subcutaneous injection; PFU—plaque-forming units.

**Figure 4 vaccines-10-00305-f004:**
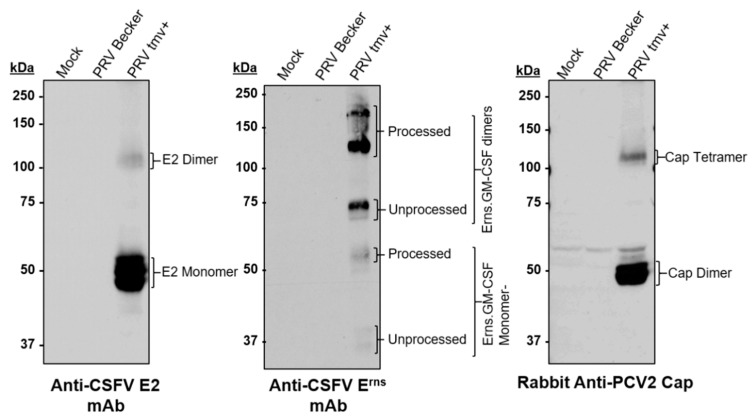
Immunoblot analysis of PRVtmv+ expressing chimeric CSFV E2, CSFV E^rns^-GM-CSF, and PCV2 cap proteins using an anti-CSFV E2 monoclonal antibody (mAbs) (left panel), an anti-CSFV E^rns^ mAbs (middle panel), and a rabbit anti-PCV2 cap Ab (right panel), respectively.

**Figure 5 vaccines-10-00305-f005:**
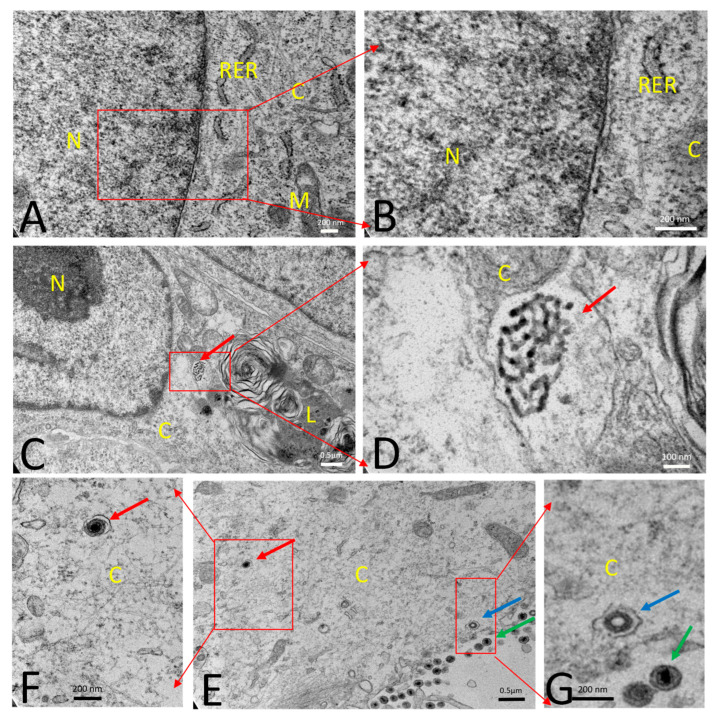
Transmission electron microscopy: (**A**,**B**) Mock infected healthy swine kidney (SK) cell with normal cellular morphology. (**C**,**D**) SK cells were infected with porcine circovirus type 2b (PCV2b) (infected at an MOI of 0.1) and fixed at 72 h post-infection (hpi). PCV2b infected cell showed accumulation of PCV2 viruses (red arrow) within the vesicle-like structures in the cytoplasm. Each PCV2 particle measured about 20 nm in diameter. (**E**–**G**) SK cells were infected with PRV wt (infected with an MOI of 5) and fixed at 12 hpi. PRV wt-infected cells showed enveloped herpesvirus (red arrow; about 200 nm in diameter) within the vesicle-like structure in the cytoplasm. The process of budding and release of several enveloped viruses were also noticed on the periphery of the cell near the plasma membrane (blue arrow). Released virus particles from the outer surface of the cells were accumulated in intercellular space (green arrow). (**B**, **D**, **F**, **G**) are magnifications of (**A**, **C**, **E**), respectively. N—Nucleus; C—Cytoplasm; RER—Rough endoplasmic reticulum; M—Mitochondria; L—Lysosomes.

**Figure 6 vaccines-10-00305-f006:**
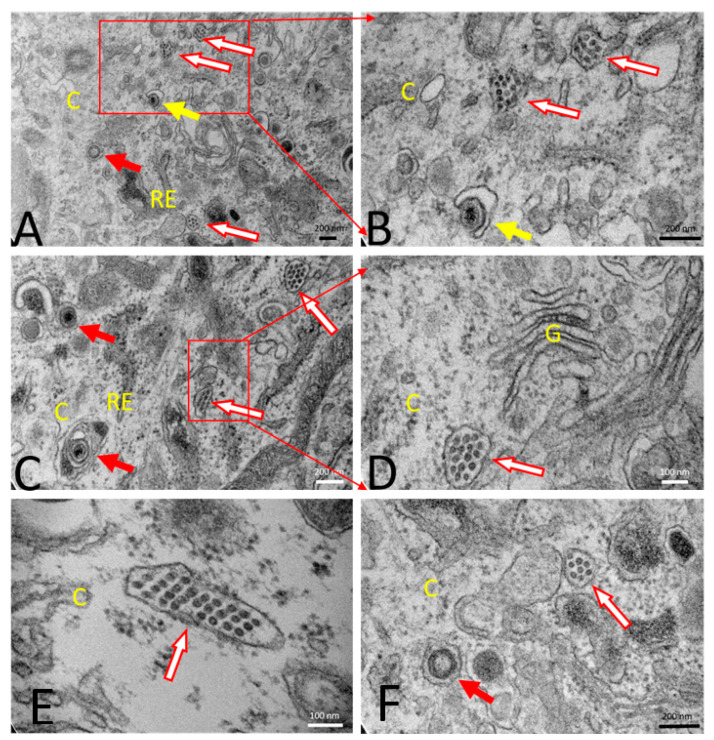
Transmission electron microscopy. (**A**) PRVtmv+ vaccine virus-infected swine kidney (SK) cells (infected at an MOI of 5) were fixed at 18 h post-infection. PRVtmv+ infected cells showed several fully enveloped 200 nm in diameter size PRVtmv+ vaccine virus particles in the exocytic vesicles in the cytoplasm (red arrows). Secondary envelopment of intracytoplasmic PRVtmv+ capsids by budding into vesicles was visible (yellow arrows). Most of the PRVtmv+ infected SK cells showed several accumulations of PCV2 virus-like particles (VLPs) within the vesicular structures in the cytoplasm (red arrow with a white fill). The PCV2-VLPs were circular, and each measured about 20 nm in diameter. (**B**) Magnification of A is given. (**C**) The presence of enveloped PRVtmv+ vaccine virus (red arrow) and PCV2 VLPs within vesicles (red arrow with a white fill) in the cytoplasm of the PRVtmv+ infected SK cells were visible. (**D**–**F**) Accumulations of PCV2-VLPs within the vesicle-like structures in the cytoplasm of the infected cells (red arrow with a white fill). C—Cytoplasm; RER—Rough endoplasmic reticulum; G—Golgi apparatus.

**Figure 7 vaccines-10-00305-f007:**
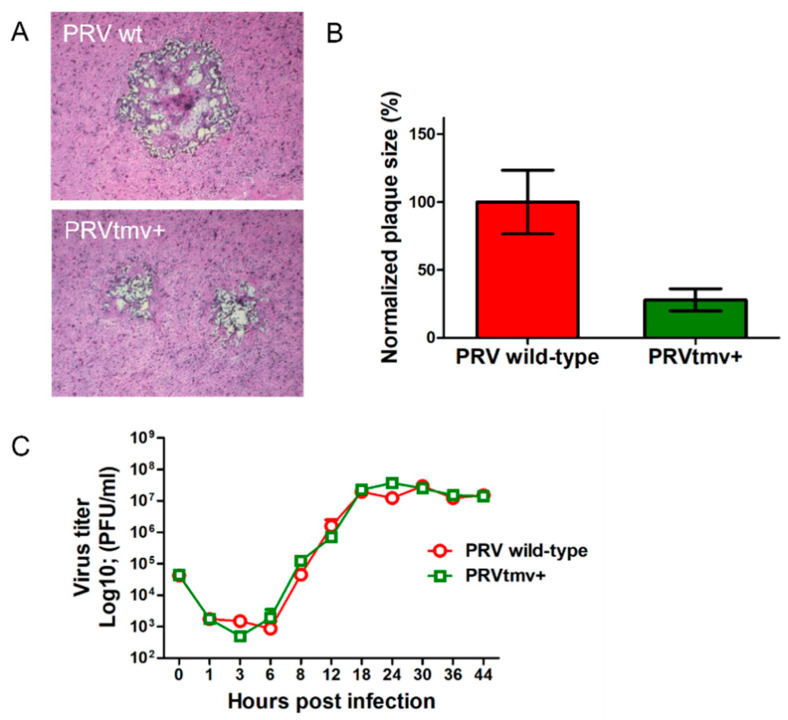
In vitro characterization of PRVtmv+. (**A**) Plaque size analysis of PRVtmv+ compared to that of PRV wt. Shown are the pictures of areas containing representative plaques of each virus. (**B**) Bar graph showing normalized average plaque size (*n* = 150) for each virus with standard deviation (SD). (**C**) One-step growth analysis of PRVtmv+ compared with PRV wt.

**Figure 8 vaccines-10-00305-f008:**
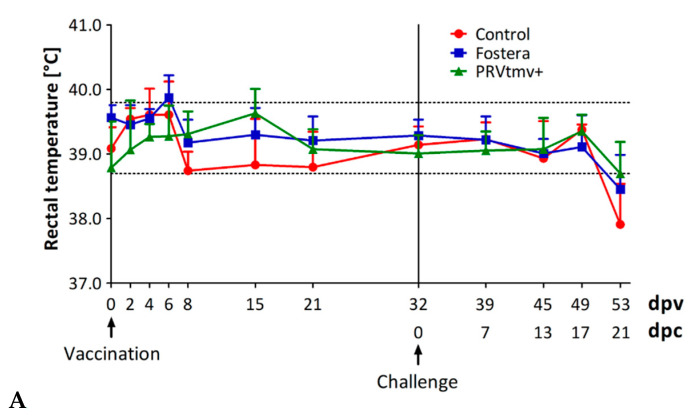
Clinical assessment. (**A**) Rectal temperature of pigs following immunization and challenge. Rectal temperature was measured using a digital thermometer on indicated days. Shown is the mean temperature of each treatment group with standard deviation (SD) (*n* = 5). (**B**) Body weight (BW) of pigs following immunization and challenge. BW was measured using a digital weight balance on indicated days. The mean BW of each treatment group with SD (*n* = 5) is shown. dpv—days post-vaccination; dpc—days post-challenge. Two-way ANOVA followed by Bonferroni post-tests to compare replicate means by row. *p* < 0.05 is considered as significant.

**Figure 9 vaccines-10-00305-f009:**
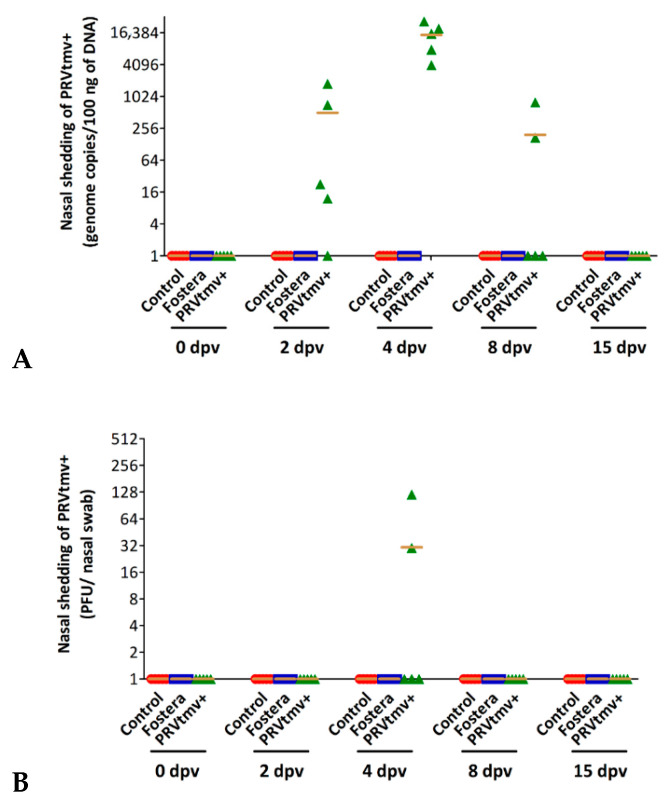
Nasal shedding of PRVtmv+ in immunized pigs assessed by qPCR and virus isolation. (**A**) DNA was isolated from nasal swab following immunization with PRVtmv+ vaccine, and PRV-qPCR was performed. PRV genome copy numbers were calculated according to the CT values of a standard curve. Shown are the mean copy numbers of PRV genome in 100 ng of DNA of two independent qPCR analyses of each animal from three vaccination groups on 0, 2, 4, 8, and 15 dpv. The dot plot graph represents mean + individual values in each group (*n* = 5). (**B**) Virus isolated from each animal’s nasal swab following the immunization with PRVtmv+ vaccine was titrated in confluent SK cells by plaque assay. Shown are the virus titers (in plaque-forming unit/mL of the nasal swab; PFU/mL) of each animal from three vaccination groups on 0, 2, 4, 8, and 15 dpv. The dot plot graph represents mean + individual values in each group (*n* = 5). dpv—days post-vaccination.

**Figure 10 vaccines-10-00305-f010:**
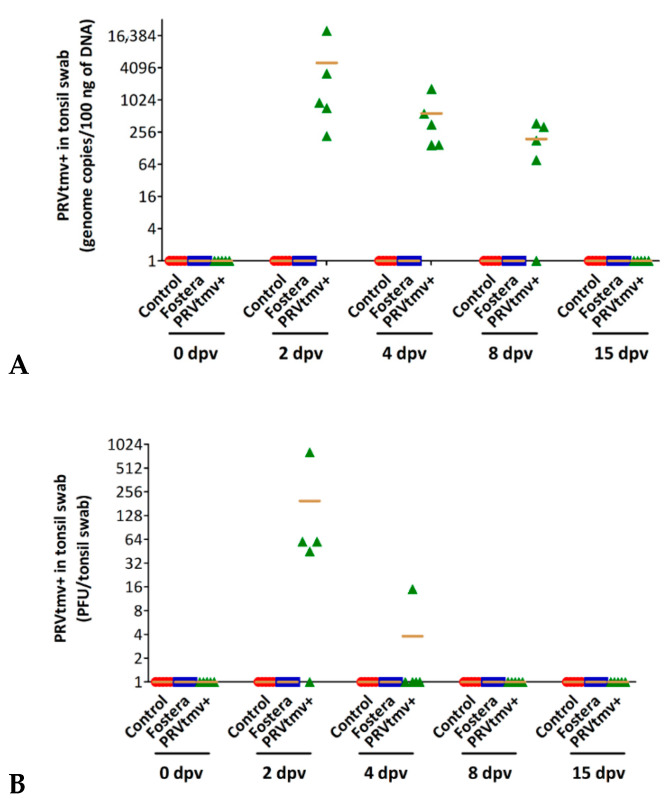
Quantification of PRVtmv+ in tonsil swab of immunized pigs assessed by qPCR and virus isolation. (**A**) DNA was isolated from tonsil swabs following immunization with PRVtmv+ vaccine, and PRV-qPCR was performed. PRV genome copy numbers were calculated according to the CT values of a standard curve. The mean copy numbers of PRV genome in 100 ng of DNA from tonsil swab of two independent qPCR analyses of each animal from three vaccination groups on 0, 2, 4, 8, and 15 dpv are shown. The dot plot graph represents mean + individual values in each group (*n* = 5). (**B**) Virus isolated from each animal’s tonsil swab following the immunization with PRVtmv+ vaccine was titrated in confluent SK cells by plaque assay. Shown are the titers (in plaque-forming unit/mL of the nasal swab; PFU/mL) of each animal from three vaccination groups on 0, 2, 4, 8, and 15 dpv. The dot plot graph represents mean + individual values in each group (*n* = 5).

**Figure 11 vaccines-10-00305-f011:**
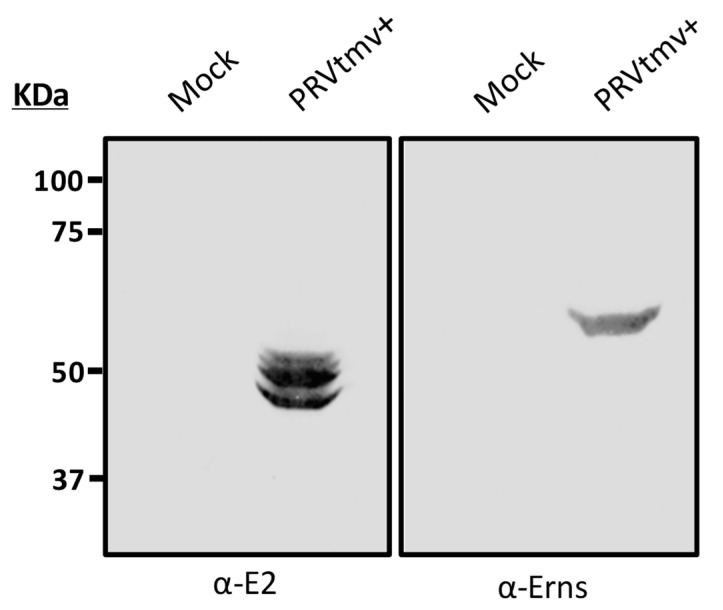
PRVtmv+ is stable in pigs. Immunoblot analysis of PRVtmv+ vaccine virus after a passage in pigs expressing chimeric CSFV E2 and E^rns^ proteins by using E2- (left panel) and E^rns^-specific (right panel) mAbs, respectively.

**Figure 12 vaccines-10-00305-f012:**
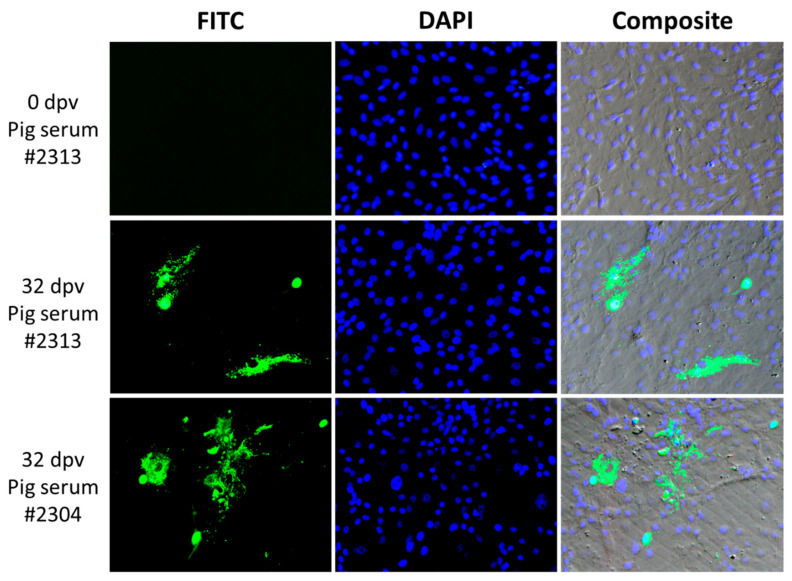
Indirect immunofluorescence assay (IIFA) for serum samples collected from PRVtmv+ immunized pigs. Swine kidney cells were transfected with PCV2b plasmid. At 72 h post-transfection, cells were fixed and IIFA were performed using serum samples collected from PRVtmv+ immunized pig on 0 day post-vaccination (dpv) (pig #2313) and 32 dpv (pig #2313 and 2304) as a primary antibody and fluorescent-labeled anti-porcine secondary antibody with DAPI nuclear stain. Positive signals were indicated by bright apple-green fluorescent signals. (Magnifications 200×).

**Figure 13 vaccines-10-00305-f013:**
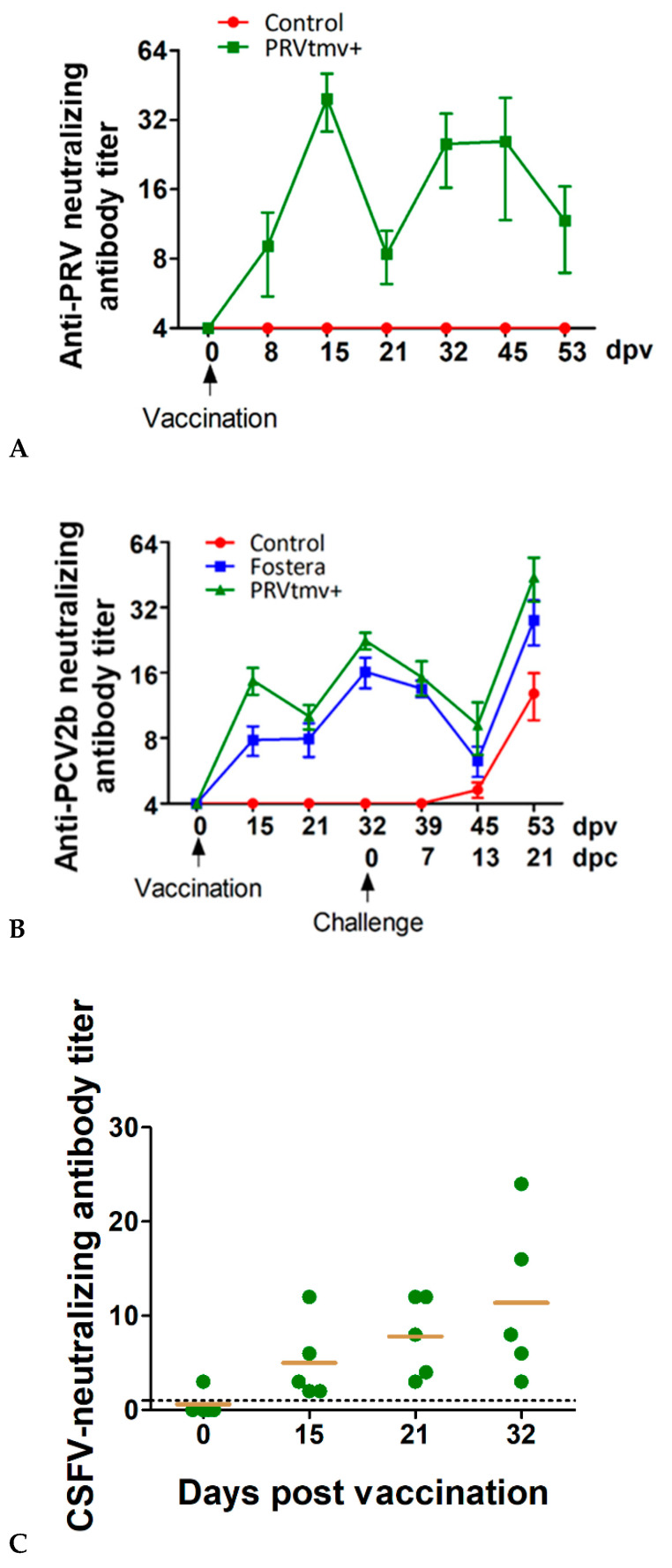
PRV-, PCV2b-, and CSFV-specific serum neutralizing (SN) antibody titer developed in pigs after PRVtmv+ vaccination. (**A**) PRV-specific SN titers. The data represent the mean + standard deviation (*n* = 5). (**B**) PCV2b-specific SN antibody titer following PRVtmv+ immunization and PCV2b challenge. PCV2b is non-cytopathic, and the viral plaques were visualized by IFA at 72 h post-inoculation using the PCV2b specific mAbs 36F1 and fluorescent-tagged secondary antibody. (**C**) CSFV-specific neutralization dose (ND_50_) SN titers following PRVtmv+ immunization. ND_50_ titers were calculated as described earlier [51] and briefly in the materials and method section. The dot plot graph shows each animal’s mean values and individual titer with standard deviation (*n* = 5). dpv—days post-vaccination; dpc—days post-challenge.

**Figure 14 vaccines-10-00305-f014:**
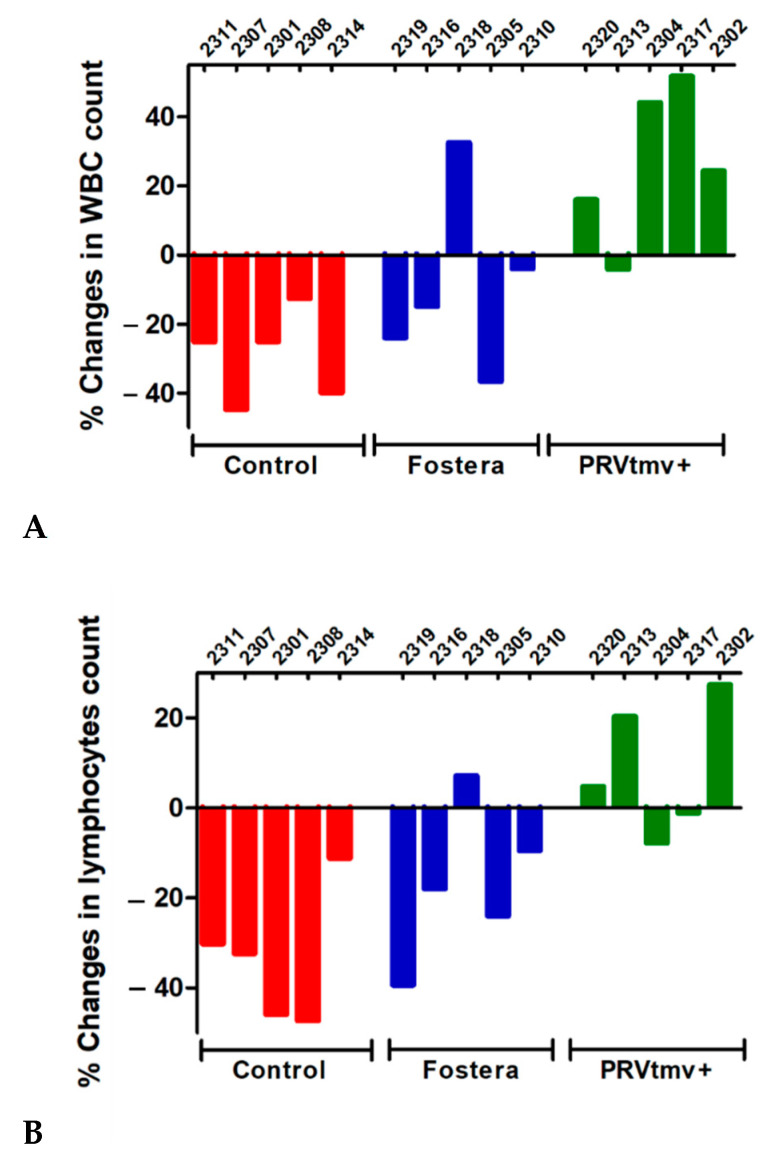
Percent changes in leukocyte and lymphocyte count following challenge in all three groups. Whole blood was collected from pigs on 32 dpv/0 dpc and 53 dpv/21 dpc, and (**A**) leukocyte and (**B**) lymphocyte counts were determined and percent changes were calculated.

**Figure 15 vaccines-10-00305-f015:**
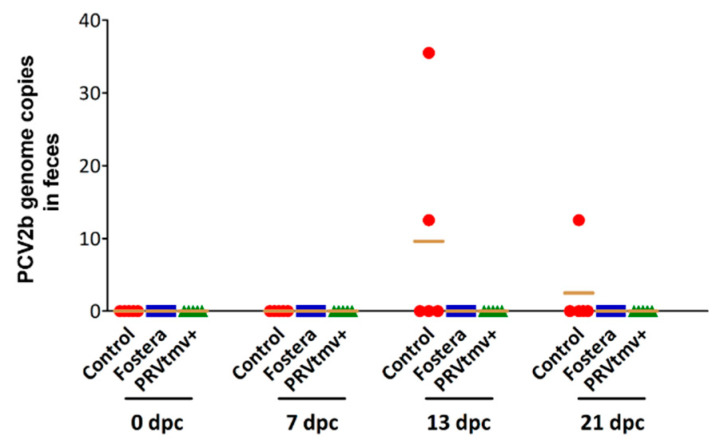
Fecal PCV2b shedding in control and vaccinated pigs following the PCV2b challenge was assessed by qPCR. DNA was isolated from fecal swab, and PCV2b qPCR was performed. PCV2b genome copy numbers were calculated according to CT-values of a standard curve. The mean copy numbers of PCV2b genome are shown in 200 ng of DNA of two independent qPCR analyses of each animal from the three vaccination groups on 0, 13, 17, and 21 dpc. The dot plot graph represents mean + individual values in each group (*n* = 5). dpc—days post-challenge.

**Figure 16 vaccines-10-00305-f016:**
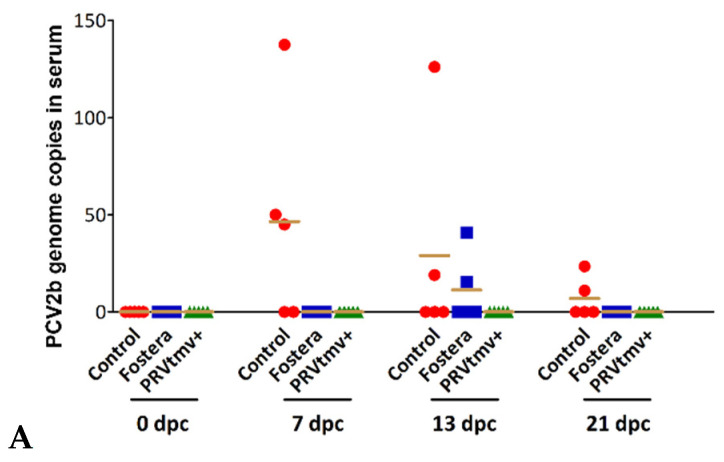
PCV2b in serum (cell-free) and PBMC (cell-associated viremia) in control and vaccinated pigs following the challenge was assessed by qPCR. DNA was isolated from serum and PBMC, and PCV2b qPCR was performed as described. PCV2b genome copy numbers were calculated according to CT-values of a standard curve. (**A**) Shown is the mean copy numbers of PCV2b genome in serum (100 ng of DNA) and (**B**) PBMC (normalized to 10^7^ cells) of two independent qPCR analysis of each animal from three vaccination group on 0, 13, 17, and 21 dpc. The dot plot graph represents mean + individual values in each group (*n* = 5). dpc—days post-challenge.

**Figure 17 vaccines-10-00305-f017:**
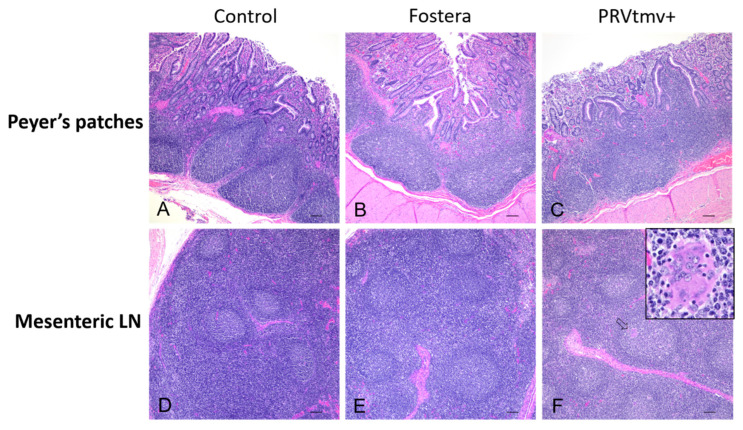
Histopathology following vaccination with Fostera or PRVtmv+ vaccines and subsequent challenge with PCV2. Control group (**A**,**D**), Fostera-vaccinated group (**B**,**E**), and PRVtmv+-vaccinated group (**C**,**F**). Overall, there are no significant histologic changes. Rare multinucleated giant cells were noted in the mesenteric lymph node of one PRVtmv+-vaccinated pig (#2302; F, arrow, and inset) H&E 100× total magnification, Bar = 100 μm.

**Figure 18 vaccines-10-00305-f018:**
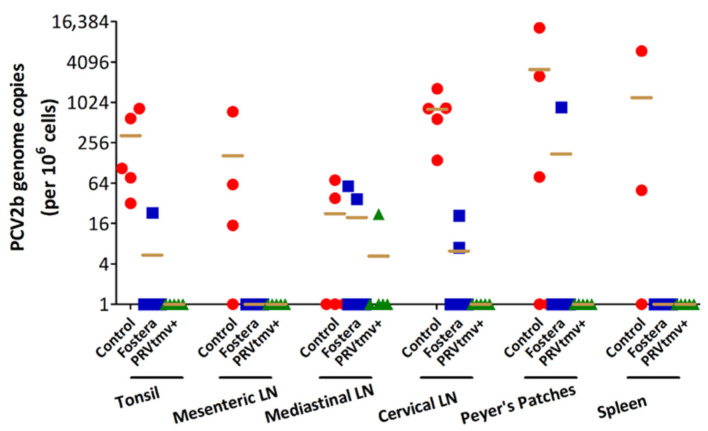
Quantification of PCV2b viral genome copies in pig tissues. DNA was isolated from 25 mg of tissues (tonsil, mesenteric LN, mediastinal LN, cervical LN, Peyer’s patch, and spleen), and PCV2b qPCR was performed. PCV2b genome copy numbers were calculated according to CT-values of a standard curve. Calculated PCV2b genome copies were normalized based on porcine house-keeping gene GAPDH, and shown are the mean copy numbers of PCV2b genome per one million cells. Two independent qPCR analyses were performed for each animal from three vaccination groups. The dot plot graph represents the mean + individual values in each group (*n* = 5).

**Figure 19 vaccines-10-00305-f019:**
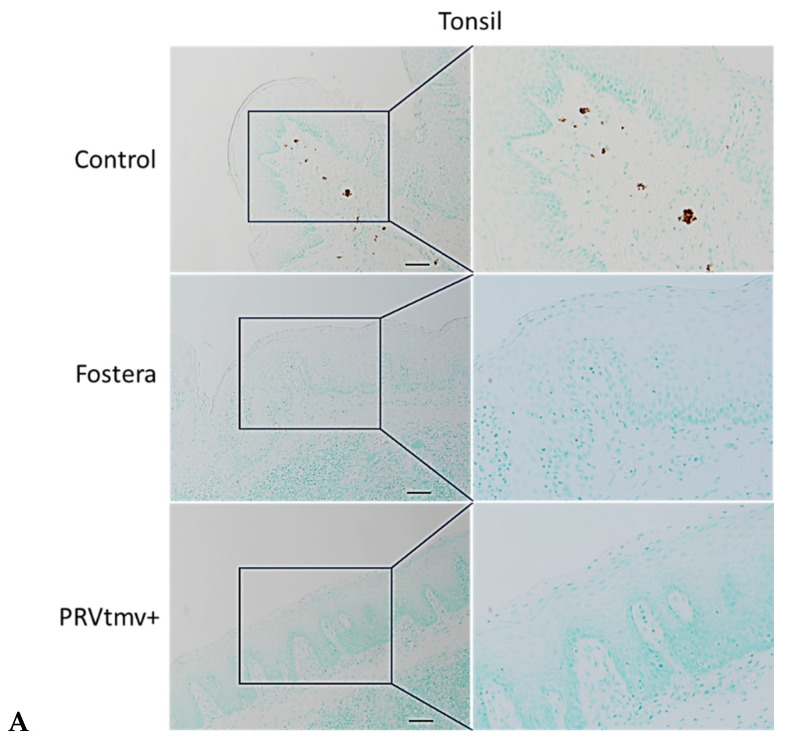
Immunohistochemistry showing the PCV2b antigen in tonsil (**A**) and intestine (Peyer’s patches) (**B**). Tissue sections were prepared, cleared, rehydrated, and subjected to immunostaining using anti-PCV2b mAb and horseradish peroxidase-labeled secondary antibodies. Immunostained sections were counterstained with 0.5% methylene green stain and examined under a microscope after mounting. PCV2 antigens in tissues were confirmed by the presence of bright-golden brown positive signals. (**A**) Note that PCV2b antigens are detected in the tonsil (**A**) and Peyer’s patch (**B**) of the control unvaccinated pig. Regardless of PRVtmv+ or Fostera vaccine group, PCV2b Cap antigen could not be detected in the tonsils of pigs. In one pig (#2310) of the Fostera group, PCV2b Cap-specific antigen in a tiny area of Peyer’s patch could be detected (**B**). Magnification 100×. Bar = 100 μm.

**Table 1 vaccines-10-00305-t001:** List of primers used to construct the gE-deleted and TK-null/deleted recombinant viruses. Restriction enzyme cleavage sites are bold and underlined. Stop codons are highlighted in bold.

Primer Name		Primer Sequence
PRV TK left flanking	Forward (F3)	5′-gc**gaattc**ccgccttatcaccgcgcccacccgctcc-3′
	Reverse (R3)	5′-cgg**ggtacc****tta**c**tca**c**tta**gcccacgaaggccgccgcgctgatgtcc-3′
PRV TK right flanking	Forward (F4)	5′-cgcgtt**aagctt**tcgctggacccggacgagcacctgcg -3′
	Reverse (R4)	5′-cca**atgcat**ggccgagcaggtgcccgcggatgc-3′

**Table 2 vaccines-10-00305-t002:** List of primers used to verify the PRV TK-deletion and PCV2 Cap chimeric gene-insertion by PCR and/or sequencing.

Primer Name		Primer Sequence
PRV TK flanking 5′ sequencing	Forward (F5)	5′-cttgctgggcgtgttgaggttcc-3′
	Forward (F6)	5′-accggcaagagcaccactgc-3′
	Reverse (R5)	5′-agaaggcgtccttgaccctgg-3′
	Reverse (R6)	5′-tcctcctcgctcaggctgc-3′
PRV TK flanking 3′ sequencing	Forward (F7)	5′-aacacgtcgcgctacctgagc-3′
	Forward (F8)	5′-cgccttcacgtcggagatggg-3′
	Reverse (R7)	5′-gccttgtacgcgccaaagaggg-3′
	Reverse (R8)	5′-ggcatggtgacgggcacg-3′
TK deletion confirmation	Forward (F9)	5′-gcgcactctgttcgacacggacacggtgg-3′
	Reverse (R9)	5′-ggccaccaccaggttgccgcc-3′
PCV Cap cloning PRV TK left/PCV2 Cap	Forward (F14)	5′-acgccgtacctgctgctgcacacg-3′
	Reverse (R14)	5′-gcatcatgtccacggcccaggaaggtgtgg-3′

**Table 3 vaccines-10-00305-t003:** List of primers, probes, and double standard gene blocks (ds-gblock as standard) used in quantitative PCR used for quantification of pseudorabies (PRV)/porcine circovirus type 2b (PCV2b) genome in samples and subsequent normalization based on glyceraldehyde 3-phosphate dehydrogenase (GAPDH) housekeeping gene.

Primer/Probe/ds-Gblock Name		Sequence
PRV (Major capsid protein)	Forward	5′-ccatccagtttgaggtgcag-3′
	Reverse	5′-cgaggcgcttgatcatgtag-3′
	Probe	5′Fam-cccgtcgcgcgcgatcatcg-3′ Tamra
	ds-gblock	5′-ctcagctacgtggccgagggcaccatccagtttgaggtgcagcagccgatgatc gcgcgcgacgggccgcacccggccgaccagcccgtgcacaactacatgatcaagc gcctcgatcgccgctccctcaacgccgc-3′
Swine Glyceraldehyde 3-phosphate dehydrogenase (GAPDH)	Forward	5′-atgacaacttcggcatcgtg-3′
	Reverse	5′-ccatccacagtcttctgggt-3′
	Probe	5′Fam-accacagtccatgccatcactgcc-3′ Tamra
	ds-gblock	5′-gcacccctggccaaggtcatccatgacaacttcggcatcgtggaaggactcatgaccacagtccatgccatcactgccacccagaagactgtggatggcccctctgggaaacgtggcgt-3′
Porcine circovirus type 2b-capsid protein (ORF2)	Forward	5′-cacagccctcacctatgacc-3′
	Reverse	5′-aaagtagcgggagtggtagg-3′
	Probe	5′Fam-cccgccataccataacccagccc-3′ Tamra
	ds-gblock	5′-atgataactttgtaacaaaggccacagccctcacctatgacccctatgtaaactacTcctcccgccataccataacccagcccttctcctaccactcccgctactttacccccaaacctgtcctagatt-3′

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
