# Peer review of "A Triple Gene-Deleted Pseudorabies Virus-Vectored Subunit PCV2b and CSFV Vaccine Protects Pigs against PCV2b Challenge and Induces Serum Neutralizing Antibody Response against CSFV"

_vaccines, 2022, doi:10.3390/vaccines10020305_

Round 1
Reviewer 1 Report
Authors constructed and assessed a PRV vectored triple mutant trivalent vaccine against PRV, PCV2b, and CSFV in pigs as per standard protocols. But the vaccine protective efficacy has been evaluated only for PCV2b against virulent virus challenge. The study is moderately novel, significant, and useful for an international research audience. Work has been meticulously planned and executed moderately well. However, a couple of clarifications or suggestions need attention from authors for the betterment of MS before publication.
Title: it is not clear that the vaccine is a vector or subunit vaccine. kindly modify the title for clarity of the audience
Introduction: Authors can limit the lengthy introduction part to some extent as it looks to be exhaustive
Materials and Methods:
1. Whether the number of pigs used in each group is statistically significant? on what basis or protocol, number of pigs to use in three groups were decided?
2. PRV triple mutant group vaccinated pigs were inoculated with Noromycin 300LA antibiotics. Why did other groups not receive such antibiotics? should be justified.
3. Authors mentioned, "At 32 days post-vaccination (dpv), animals of all three groups were challenged with PCV2b IN with a total of 1.6 × 104 PFU (8 × 103 PFU/nostril) and SC with 6.75 × 103 PFUs".
Authors selected vaccinated pigs along with controls for the challenge at 32 dpv. The reason behind 32dpv for the challenge to be discussed as it is common to challenge at 28 dpv for most of the vaccines.
4. The dose of challenge virus shown here seems to be low (1.6 × 104 PFU) given IN and S/c. What is the basis of such a low dose used in the challenge? if a typographical mistake, it should be corrected.
Results:
The safety of the vaccine under the study is to be clearly mentioned as it also reduced the weight gain (only 3% vs control 15%) at 6 dpv
Author Response
- Comment: Title: it is not clear that the vaccine is a vector or subunit vaccine. kindly modify the title for clarity of the audience
Response: We have modified the title slightly to clarify the reviewer's comment to "A Triple Gene-deleted Pseudorabies Virus-vectored Subunit PCV2b and CSFV Vaccine Protects pigs against PCV2b challenge and induces serum neutralizing antibody response against CSFV".
- Comment: Introduction: Authors can limit the lengthy introduction part to some extent as it looks to be exhaustive.
Response: We have revised and shortened the introduction.
- Comment: Whether the number of pigs used in each group is statistically significant? on what basis or protocol, number of pigs to use in three groups were decided?
Response: The sample size was determined based on a power analysis; this is now included on page 14, Line no.: 544-551.
- Comment: PRV triple mutant group vaccinated pigs were inoculated with Noromycin 300LA antibiotics. Why did other groups not receive such antibiotics? should be justified.
Response: On the day of immunization (day 0 post immunization), pigs in PRVtmv+ group was immunized with live PRV-tmv.PCV2/CSFV virus vaccine. As these pigs were given with live virus vaccine, prophylactic Noromycin 300LA antibiotic was administered to avoid secondary bacterial infections. On the other hand, i) control group pigs were mock inoculated with cell culture supernatant and ii) Fostera group pigs were given with inactivated Fostera® Gold PCV which does not contain any infectious agent to cause any secondary bacterial complications.
- Comment: Authors mentioned, "At 32 days post-vaccination (dpv), animals of all three groups were challenged with PCV2b IN with a total of 1.6 × 104 PFU (8 × 103 PFU/nostril) and SC with 6.75 × 103 PFUs". Authors selected vaccinated pigs along with controls for the challenge at 32 dpv. The reason behind 32dpv for the challenge to be discussed as it is common to challenge at 28 dpv for most of the vaccines.
Response: We agree with reviewer's comment regarding day of PCV2b challenge post-vaccination. In most of the studies, PCV2 challenge were performed at 4-weeks post vaccination (on day 28). Due to technical reasons (Covid restrictions), we have decided to challenge the pigs on day 32 post-vaccination. Further, four days difference should not have made much difference in the outcome of the findings.
- Comment: The dose of challenge virus shown here seems to be low (1.6 × 104 PFU) given IN and S/c. What is the basis of such a low dose used in the challenge? if a typographical mistake, it should be corrected.
Response: As PCV2b is slow replicating virus, does not replicate to high titer like other viruses. The normal PCV2b yield in the cell culture is around 104 plaque-forming unit (PFU)/ml. Previous studies revealed that 104 PFU/ml of PCV2b is the infectious dose for pigs (Vet Microbiol. 2011 Apr 21;149(1-2):91-8). Further, several vaccine efficacy studies showed that following immunization, the standard PCV2 challenge dose in pigs is in the range of 104 PFU per animal (Clin Vaccine Immunol. 2011, 18(8): 1261–1268; Front. Microbiol. 2018, https://doi.org/10.3389/ fmicb.2018.00455; Virology Journal, 2015, 12(113)). Based on these previous studies, we have decided to use the dose of PCV2b (1.6 x 104 PFU - intranasal and 6.75 x 103 PFU – subcutaneous per pig) for our challenge study.
- Comment: The safety of the vaccine under the study is to be clearly mentioned as it also reduced the weight gain (only 3% vs control 15%) at 6 dpv
Response: Transient body weight reduction following immunization is very common and acceptable in pig production. Here we want to mention that the Fostera group pigs showed about 1.4% reduction in body weight on day 6 post vaccination. But, on the other hand, both control and PRVtmv+ immunized pigs showed gain in body weight on day 6 post vaccination. The percent of weight gain were 3% and 15% in PRVtmv+ and control group, respectively. These differences in body weights were not statistically significant (p>0.05). None of the pigs from our vaccine group developed any adverse reactions following immunization. So, we consider our vaccine is safe for pigs as mentioned in the manuscript.
Reviewer 2 Report
Comments for the authors
- L200-202: add reference
- L321: Figure 1
- L440: body weight (BW)
- L446, 676-678: BW instead of body weight
- L988-990: refresh the sentence ‘’In many East European and Asian countries, including China, the coinfection rates of PRV with PRRSV, PCV2, CSFV are high in intensive pig farms.’’
- L1041-1082: please add all appropriate information to the following: Supplementary, Materials, Funding, Institutional Review Board Statement, Informed Consent Statement, Data Availability Statement, Conflicts of Interest
Author Response
- Comment: L200-202: add reference
Response: Reference have been added as suggested (J Virol 2000, 74, 9553-9561) Line no.: 175.
- Comment: L321: Figure 1
Response: Modified as Suggested. Line no.: 292.
- Comment: L440: body weight (BW)
Response: Modified as Suggested. Line no.: 414.
- Comment: L446, 676-678: BW instead of body weight
Response: Modified as Suggested. Line no.: 419, 660-661.
- Comment: L988-990: refresh the sentence "In many East European and Asian countries, including China, the coinfection rates of PRV with PRRSV, PCV2, CSFV are high in intensive pig farms."
Response: Modified as Suggested. Line no.: 965-966.
- Comment: L1041-1082: please add all appropriate information to the following: Supplementary, Materials, Funding, Institutional Review Board Statement, Informed Consent Statement, Data Availability Statement, Conflicts of Interest
Response: Supplementary, Materials, Funding, Institutional Review Board Statement, Conflicts of Interest details have been added. Line no.: 1021-1053.
Reviewer 3 Report
The manuscript entitled “A Triple Gene-deleted Pseudorabies Virus Vectored PCV2b 2 and CSFV Subunit Vaccine Protects pigs against PCV2b challenge and induces serum neutralizing antibody response 4 against CSFV” describes the development of a vaccine to simultaneously protect pigs against classical swine fever, pseudorabies and PCV2b associated diseases by strategically assembling relevant immunogenic gens in a genetically modified pseudorabies virus vector.
The article contains very detailed information in its methodology section, which is certainly very helpful for other researchers who might need to reproduce the experiments or to the ones who desire to develop similar experiments using this article as a reference. Nevertheless, there are some points of improvement that should be considered before publication, as follows.
Although titles that state the general conclusion of the study are more appealing to readers, the proposed title is too long. Consider a shorter version.
Overall, the introduction is excessively long, making it very difficult to follow and to catch what is really important to understand the following methods and results. I suggest reducing the introduction to about five paragraphs. Some general details about the three viruses can be omitted by orienting the reader directly to the main references that covers the general information of each virus.
The discussion is superficial and repeats most of the information that had been provided in the introduction. Consider re-writing the whole discussion section focusing on explaining the results, limitations, unexpected outcomes and so on.
Other specific comments:
Page 1 and 2, lines 35 to 45: one of the most important consequences of PCV2 infection is immunosuppression and it needs to be acknowledged in introduction.
Page 2, line 58: What does PRV stands for? As it might not be obvious to the readers I suggest to write the whole words before abbreviating them.
Page 18, lines 670 - 671: Was the difference in the percent of weight gain statistically tested? If not, that would be acknowledged in the sentence.
Figures 10 and 16: Check the formatting of the figures, as the “B”s for the second graphs seem misplaced.
Page 11, lines 434 to 443: What was the vehicle of the proposed vaccine and the PCV2b suspension? How was the route of vaccine administration determined? What kind of adjuvant are you considering to test in the next experiments?
Figure 17F: The insert does not provide enough details assure that the cell is a multinucleated cell. I suggest using another figure with higher quality and a bigger insert.
Page 30, lines 959 to 962: How do the authors explain the leukopenia and lymphopenia in pigs vaccinated with Fostera and the increase in leucocites and lymphocites numbers in pigs vaccinated with the proposed vaccine?
Pages 30 and 31, lines 971 to 1006: This part of the discussion is repeating the information already provided in introduction.
Author Response
- Comment: Although titles that state the general conclusion of the study are more appealing to readers, the proposed title is too long. Consider a shorter version.
Response: We think the reviewer is referring to the title the editorial office has mistakenly used in the correspondence. Our title is appropriate as written.
- Comment: Overall, the introduction is excessively long, making it very difficult to follow and to catch what is really important to understand the following methods and results. I suggest reducing the introduction to about five paragraphs. Some general details about the three viruses can be omitted by orienting the reader directly to the main references that cover the general information of each virus.
Response: We have addressed this issue above (Reviewer 1).
- Comment: The discussion is superficial and repeats most of the information that had been provided in the introduction. Consider re-writing the whole discussion section focusing on explaining the results, limitations, unexpected outcomes, and so on.
Response: We have modified the discussions.
- Comment: Page 1 and 2, lines 35 to 45: one of the most important consequences of PCV2 infection is immunosuppression and it needs to be acknowledged in introduction.
Response: As per suggestion, we have added (Line no.: 38-42) details regarding PCV2 induced immunosuppression.
- Comment: Page 2, line 58: What does PRV stands for? As it might not be obvious to the readers I suggest to write the whole words before abbreviating them.
Response: Thank you for the comments. We wrote whole words before abbreviating them. Pseudorabies virus have been abbreviated to PRV in line no.: 47-48 before using the abbreviation in line no.: 62.
- Comment: Page 18, lines 670 - 671: Was the difference in the percent of weight gain statistically tested? If not, that would be acknowledged in the sentence.
Response: We have used Two–way ANOVA followed by Bonferroni post–tests to compare replicate means by row. p < 0.05 is considered as significant. The body weight difference between the groups were not statistically significant (p>0.05). Line no.: 552-555.
- Comment: Figures 10 and 16: Check the formatting of the figures, as the "B" s for the second graphs seem misplaced.
Response: Modified and reformatted as suggested.
- Comment: Page 11, lines 434 to 443: What was the vehicle of the proposed vaccine and the PCV2b suspension? How was the route of vaccine administration determined? What kind of adjuvant are you considering to test in the next experiments?
Response: We have used live Pseudorabies Virus Vectored PCV2b and CSFV Subunit Vaccine for the current study. The live vectored virus vaccines does not require any adjuvants for high immunogenicity. It is the main advantage of using live vectored virus vaccines. PCV2b virus stock was grown in swine kidney cells. So, we have used the 0.2µm filtered virus suspended in the cell culture medium for the challenge study following immunization. Regarding the route of administration for vaccines, i) for our vaccine i.e. PRVtmv+, we have immunized the pigs both intranasal and subcutaneous. We have used PRVtmv as a vector, which replicates well in the upper respiratory tract epithelium and tonsil following natural infection and stimulate optimum neutralizing antibody response. Subcutaneous vaccine administration results in better stimulation of immune cells including dendritic cells. To obtain robust immune response, we have decided and chosen both intranasal and subcutaneous route for our vaccine. ii) Inactivated Fostera vaccine was given via intramuscular injection as per manufacturer's recommendations. In the current study, we have not used any chemical adjuvants as mentioned earlier, instead, we have used immune-adjuvant – which is Granulocyte-macrophage colony-stimulating factor (GM-CSF). We have fused GM-CSF with CSFV Erns and expressed as secreted form of CSFV Erns-GM-CSF fusion protein. GM-CSF is an attractive adjuvant for live vectored vaccines on account of its ability to recruit antigen-presenting cells (APCs) to the site of antigen synthesis as well as its ability to stimulate the maturation of dendritic cells. In future, we would improve this vaccine by rationale combination of immune-adjuvants like Cytokines (IL-2, IL-12), toll-like receptors, etc.
- Comment: Figure 17F: The insert does not provide enough details assure that the cell is a multinucleated cell. I suggest using another figure with higher quality and a bigger insert.
Response: We have inserted a better and higher quality inset Figure 17F.
- Comment: Page 30, lines 959 to 962: How do the authors explain the leukopenia and lymphopenia in pigs vaccinated with Fostera and the increase in leucocites and lymphocites numbers in pigs vaccinated with the proposed vaccine?
Response: We have determined leukopenia and lymphopenia by comparing leukocyte/lymphocyte counts on day 21 post- challenge with that of day 0-post challange (prior to or on the day of challenge). About 20-25% reduction in leukocyte/lymphocyte count between 0 dpc and 21 dpc is considered as leukopenia and lymphopenia. Similarly, about 25% increase reflected as leukocytosis and lymphocytosis. Our PRVtmv+ resulted in leukocytosis and lymphocytosis in most of the pigs as mentioned in line no.: 793-801. As the PCV2b is immunosuppressive in nature, it causes depletion of leukocyte/lymphocyte (Vet Immunol Immunopathol. 2003, 92(3-4):97-111) as we observed in control-PCV2b challenged pigs. Partial protection of pigs against PCV2b by Fostera vaccine (as evidenced by incidence of cell-free and peripheral blood mononuclear cells-associated PCV2b viremia in serum and presence of PCV2b genome copies in lymphoid tissues) could be the reason for leukopenia and lymphopenia of some of the inactivated Fostera vaccine immunized pigs. On the contrary, our PRVtmv+ prevented the occurrence of PCV2b viremia, and replication of PCV2b in lymphoid tissues indicating a better lymphoproliferative immune-stimulation upon challenge. This could be the reasons for leukocytosis and lymphocytosis in our vaccine group.
- Comment: Pages 30 and 31, lines 971 to 1006: This part of the discussion is repeating the information already provided in introduction.
Response: We have shortened the introduction and now they are not repeated in the discussion.
Round 2
Reviewer 3 Report
Dear authors,
Although improvements have been made in comparison to the first version of the manuscript, I still have concerns about introduction and discussion, as follows.
Both sections are extremely important to make the study comprehensible for the audience and, hence, improvements on them would make the final article to gain more audience.
- Despite the authors' response, the introduction, instead of being shortened, gained about 45 lines, making it even more difficult to follow than the first version.
- Discussion remains superficial if we consider the amount of data the authors collect and questions that might raise in the readers, such as the choice of vaccination route (subcutaneous, intramuscular or intranasal), the possible explanation for leukopenia in pigs vaccinated with the prototype vaccination, and so on.
Author Response
Comments:
Although improvements have been made in comparison to the first version of the manuscript, I still have concerns about introduction and discussion, as follows.
Both sections are extremely important to make the study comprehensible for the audience and, hence, improvements on them would make the final article to gain more audience.
- Despite the authors' response, the introduction, instead of being shortened, gained about 45 lines, making it even more difficult to follow than the first version.
Response: The statement is not correct. We have shortened the “introduction” section considerable. It now extends to line 149 which includes also the abstract.
- Discussion remains superficial if we consider the amount of data the authors collect and questions that might raise in the readers, such as the choice of vaccination route (subcutaneous, intramuscular or intranasal), the possible explanation for leukopenia in pigs vaccinated with the prototype vaccination, and so on.
Response: We have now addressed everything the reviewer is asking for and added another whole page.
I hope that the revised version will now be acceptable to the third reviewer for publication.